# CMDPO: Centered Mirror Descent Policy Optimization for Stable and Efficient Reinforcement Learning

## Abstract

Large language models (LLMs) have shown strong performance in diverse tasks but require post-training alignment, where reinforcement learning plays a key role. Existing methods such as proximal policy optimization (PPO) and direct preference optimization (DPO) suffer from limitations like high computational overhead and overfitting. Although group relative policy optimization (GRPO) addresses some of these issues, its reliance on weighted negative log-likelihood lacks theoretical convergence guarantees. Furthermore, mirror descent policy optimization (MDPO), while more stable, requires computationally expensive partition function estimation. To overcome these challenges, this study introduces centered mirror descent policy optimization (CMDPO), a policy optimization framework that eliminates the need for explicit partition function estimation through group centering. CMDPO ensures unbiased and consistent estimates with strong theoretical guarantees. Optionally, we add two lightweight utilities for improved stability: dynamic reward weighting to balance heterogeneous rewards and token-level discriminative learning to reduce shared-segment dominance. Comprehensive experiments across multiple benchmark datasets demonstrate the effectiveness and robustness of CMDPO, which is further proven theoretically as a promising approach for LLMs' post-training. The code is accessible at `https://anonymous.4open.science/r/CMDPO-0C26`.

## 1 Introduction

Large language models (LLMs) have demonstrated remarkable performance in mathematical reasoning, multimodal understanding, etc. However, LLMs require post-training Carta et al. (2023); Sun et al. (2024); Zang et al. (2025) to better align with specific tasks or user preferences. Reinforcement learning (RL) plays a critical role in this stage.

The core objective of RL is to learn a policy that maximizes the expected long-term return in complex environments. Proximal policy optimization (PPO) Schulman et al. (2017) has been widely adopted among the many policy optimization methods due to its strong empirical performance. However, PPO suffers from training instability and high computational overhead. To address these limitations, Direct preference optimization (DPO) Rafailov et al. (2023) simplifies PPO by leveraging a closed-form relationship between the optimal policy and a reward model under a KL divergence constraint, significantly reducing resource consumption. Despite its efficiency, DPO heavily relies on human-annotated data and is prone to overfitting, leading to suboptimal generalization.

Group relative policy optimization (GRPO) Guo et al. (2025) was recently proposed to balance efficiency and robustness. GRPO mitigates the memory bottleneck of PPO through group reward normalization while retaining the reward model, thereby avoiding the overfitting issues observed in DPO. While GRPO achieves strong performance across various domains, its optimization objective still relies on the weighted negative log-likelihood of sampled responses. It lacks theoretical guarantees for convergence, leading to potential instability during training.

To address these concerns, the Kimi team introduced mirror descent policy optimization (MDPO) Team et al. (2025a), which constructs an optimization objective with a unique optimal solution based on the closed-form optimal policy. MDPO enhances the stability and theoretical rigor in

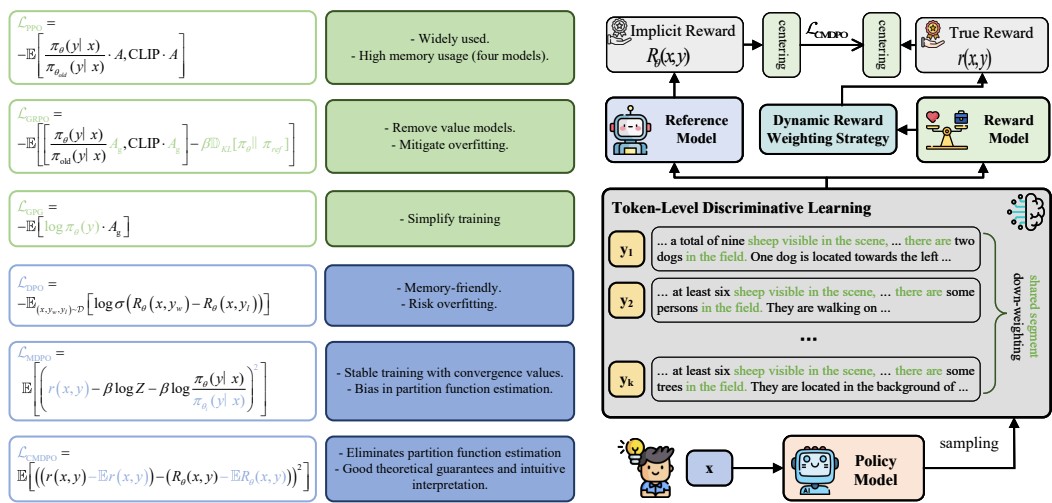

Figure 1: Comparison of policy optimization methods and the proposed framework. Here, $A$ represents the advantage calculated by the value model, and $A_g$ represents the advantage calculated by group normalization. $\text{CLIP} = \text{clip}\left(\frac{\pi_\theta(y|x)}{\pi_{old}(y|x)}, 1 - \epsilon, 1 + \epsilon\right)$ denotes the clipping function.

policy learning. However, one major limitation of MDPO lies in estimating the partition function $\log Z$. Estimating $\log Z$ requires integrating over the entire action space, which is a computationally infeasible task in high-dimensional settings such as language generation. Yet, this term is critical for maintaining the policy's normalization and theoretical guarantees. In real-world settings, $\log Z$ is approximated using limited samples by group reward averaging. Due to sampling constraints and limited computational resources, such approximations introduce significant bias, degrading model performance. We summarize the formulas and characteristics of the previous methods in Figure 1.

A natural and important question arises: Can we design a new policy optimization method that preserves the theoretical advantages of MDPO while eliminating the need for explicit estimation of the partition function $\log Z$?

The study proposes centered mirror descent policy optimization (CMDPO), a policy optimization framework. CMDPO retains the convergence properties of MDPO while introducing a group centering mechanism. Specifically, it subtracts the sample mean from explicit and the implicit (logit-derived) rewards, effectively canceling the intractable partition term $\log Z$ in the optimization objective. This centering strategy brings several key advantages: (1) It removes the dependency on the uncomputable $\log Z$. (2) It provides an unbiased and consistent estimate. (3) It ensures a unique optimal solution with strong theoretical guarantees and interpretable statistical properties.

Beyond the core algorithm, we further explore two optional and method-agnostic heuristics that improve practical training dynamics:

- **Dynamic Reward Weighting**: An adaptive scheme that rebalances task rewards and format rewards based on the convergence behavior of the task-specific signal, thereby enabling more effective utilization of the multidimensional true reward space.

- **Token-level Discriminative Learning**: A mechanism that identifies shared, non-discriminative token fragments across sampled responses and down-weights them, enabling finer-grained optimization and more accurate implicit reward attribution.

Our main contributions are summarized as follows:

- CMDPO is proposed as a theoretically grounded policy optimization algorithm that eliminates the need for estimating $\log Z$.

- A dynamic reward weighting mechanism is designed toadaptively balance task-specific and format-specific signals during training. Further, a token-level discriminative learning approach is introduced to enhance the model's ability to identify high-quality outputs.

- Comprehensive empirical evaluations across multiple benchmark datasets are conducted, verifying the effectiveness and generality of the proposed approach.

## 2 RELATED WORKS

### 2.1 LARGE MODEL REASONING

Recent advances in large language models (LLMs) Zhao et al. (2023); Minaee et al. (2024) and multimodal LLMs (MLLMs) Wang et al. (2024b); Liu et al. (2024) have increasingly focused on enabling human-like step-by-step reasoning. In the domain of LLMs, methods such as chain-of-thought (CoT) prompting Wei et al. (2022); Zheng et al. (2024), tree-of-thought Yao et al. (2023), graph-of-thought Besta et al. (2024), and Monte Carlo tree search have been proposed to elicit structured reasoning paths by generating explicit intermediate steps Trinh et al. (2024); Wan et al. (2024); Wang et al. (2024a).

To further improve reasoning capabilities, some works construct high-quality reasoning datasets for supervised fine-tuning Muennighoff et al. (2025). Others explore training with multiple reasoning paths using algorithms like DPO or PPO Chen et al. (2025); Lai et al. (2024b). Step-level process supervision has also been widely adopted to refine the reasoning process more granularly Trinh et al. (2024); Wan et al. (2024); Wang et al. (2024a); Chen et al. (2025). However, these approaches often require extensive human annotation or intensive computational resources, limiting their scalability.

Recently, DeepSeek-R1 Guo et al. (2025) has presented a promising alternative by leveraging large-scale reinforcement learning with format-specific and outcome-oriented reward functions. This method encourages self-emergent CoT reasoning without relying on manually curated datasets or complex multi-stage training pipelines, significantly reducing labor and computational costs while achieving strong performance on challenging reasoning tasks.

Inspired by this progress, researchers have applied similar strategies to MLLMs, leading to notable improvements in visual reasoning Liu et al. (2025d); Zhang et al. (2025).

### 2.2 REINFORCEMENT LEARNING

Reinforcement learning (RL) has achieved significant progress in solving sequential decision-making problems, where policy gradient methods Sutton et al. (1998) form a fundamental framework for optimizing stochastic policies. As one of the earliest approaches, REINFORCE Williams (1992) established a foundational structure for policy updates but suffers from high-variance gradient estimation, which limits its scalability.

Subsequent studies have proposed various enhancements to improve training stability. For instance, trust region policy optimization (TRPO) Schulman et al. (2015) introduces second-order approximations and constraint mechanisms to improve monotonic policy. Building upon TRPO, Proximal policy optimization (PPO) Schulman et al. (2017) further simplifies the optimization by employing a clipped surrogate objective while preserving performance stability. In recent years, many works have improved PPO by refining implementation details. Despite its wide adoption in applications such as language model fine-tuning and robotic control, PPO often involves loading multiple models during training, leading to high computational costs and complex training procedures.

Several emerging methods have been proposed to address these challenges and simplify the advantage estimation used in PPO, which typically relies on value networks. For example, ReMax Li et al. (2024) replaces the baseline with greedy sampled reward estimates, RLOO Ahmadian et al. (2024) uses leave-one-out average rewards as the baseline, and GRPO Guo et al. (2025) introduces group reward normalization. Moreover, some approaches substitute reward models with rule-based outcome-oriented reward functions, effectively alleviating memory bottlenecks. The group policy gradient (GPG) Chu et al. (2025) method removes Kullback-Leibler (KL) divergence constraints and proposes eliminating clipping in GRPO and PPO to restore exploration capabilities. Interest-

ingly, GPG and related approaches Chu et al. (2025); Yu et al. (2025); Liu et al. (2025b) converge on discarding KL regularization during policy optimization.

Despite their promising results, these methods generally lack formal convergence guarantees and may exhibit instability during training. Furthermore, removing KL constraints increases the risk of catastrophic forgetting. The Kimi team proposed mirror descent policy optimization (MDPO) Team et al. (2025a) to tackle this issue, which derives an optimal policy update based on a closed-form relationship between KL divergence and reward. MDPO defines a squared loss with a well-defined minimum, ensuring clear optimization objectives. However, due to the estimation challenge of the log-partition function $\log Z$, MDPO faces a trade-off between efficiency and accuracy.

This study eliminates the need for estimating $\log Z$ in MDPO through a group centering mechanism, achieving a strictly theoretically grounded optimal solution.

## 3 METHOD

Figure 1 illustrates the proposed CMDPO framework. We first review the foundation of MDPO, then introduce the centered reward formulation that avoids estimating $\log Z$. After presenting the theoretical properties of CMDPO, we describe two optional training enhancements, namely a dynamic reward weighting mechanism and a token level discriminative learning scheme.

### 3.1 PRELIMINARIES

The online policy mirror descent algorithm is widely employed in reinforcement learning,

$$\max_\theta \mathbb{E}_{(y,z)\sim\pi_\theta}\left[r\left(x,y\right)\right] - \beta\,\mathrm{KL}\left(\pi_\theta(y\,|\,x)\|\pi_{\theta_i}(y\,|\,x)\right) \tag{1}$$

This formulation has been shown to admit a closed-form solution,

$$\pi^*(y\mid x) = \pi_{\theta_i}(y\mid x)\exp(r^*(x,y)/\beta)/Z \tag{2}$$

where $Z = \sum_y \pi_{\theta_i}(y\mid x)\exp(r(x,y)/\beta)$ is a partition function. From this, we can further derive:

$$r^*(x,y) - \beta\log Z = \beta\log\frac{\pi^*(y\mid x)}{\pi_{\theta_i}(y\mid x)} \tag{3}$$

Motivated by the closed-form solution, the mirror descent policy optimization (MDPO) method was proposed.

$$\mathcal{L}_{\mathrm{MDPO}} = \mathbb{E}_{x,y\sim\pi_{\theta_i}}\left[\left(r\left(x,y\right) - \beta\log Z - \beta\log\frac{\pi_\theta(y\mid x)}{\pi_{\theta_i}(y\mid x)}\right)^2\right] \tag{4}$$

Although the Equation (4) admits a unique optimal solution, estimating $\log Z$ is computationally expensive. In practice, MDPO approximates $\log Z$ using the average reward. This approximation is reasonable when the group number $k \to \infty$ (proof in Appendix A.1). However, due to computational constraints, the group number is typically very limited (usually $k \leq 8$), leading to significant estimation bias.

### 3.2 CENTERED MIRROR DESCENT POLICY OPTIMIZATION

In DPO, the implicit reward $R_\theta(x,y) = \beta(\log\frac{\pi_\theta(y|x)}{\pi_{ref}(y|x)} + \log Z)$ is substituted into Bradley-Terry model to eliminate the estimation of $\log Z$. However, the method is not suitable for squared loss. Inspired by the group reward normalization in GRPO, we introduce a group centering strategy, leveraging the following alternative loss:

$$\mathcal{L}_{\mathrm{CMDPO}}(\theta) = \mathbb{E}_{x,y\sim\pi_\theta}\left[\left((r\left(x,y\right) - \mathbb{E}_y r\left(x,y\right)) - (R_\theta(x,y) - \mathbb{E}_y R_\theta(x,y))\right)^2\right] \tag{5}$$

where $R(x,y) - \mathbb{E}_y R(x,y) = \beta(log\frac{\pi_\theta(y|x)}{\pi_{ref}(y|x)} - \mathbb{E}_y log\frac{\pi_\theta(y|x)}{\pi_{ref}(y|x)})$ eliminates the estimation of $\log Z$. Specifically, for a set of samples $(x,y_i)$, both the differences of true rewards and implicit rewards concerning their expectations exhibit a zero-sum property, i.e., $\sum_y\left(r(x,y) - \mathbb{E}_y r(x,y)\right) = 0, \sum_y\left(R_\theta(x,y) - \mathbb{E}_y R_\theta(x,y)\right) = 0$.

The loss can also be reformulated as follows:

$$
\begin{aligned}
\mathcal{L}_{\text{CMDPO}}(\theta) = & \mathbb{E}_{x,y\sim\pi_\theta}[(r(x,y) - \mathbb{E}_y r(x,y))^2 \\
& + \underbrace{(R_\theta(x,y) - \mathbb{E}_y R_\theta(x,y))^2}_{\text{Variance}} - 2\underbrace{(R_\theta(x,y) - \mathbb{E}_y R_\theta(x,y))(r(x,y) - \mathbb{E}_y r(x,y))}_{\text{Covariance}}]
\end{aligned}
\tag{6}
$$

It can be observed that the loss minimizes the variance of the implicit reward while maximizing the covariance between the implicit and true rewards. Besides, the loss function is guaranteed a unique optimum and an unbiased, consistent estimator (proof in Appendix A.2 and A.3).

**Theorem 1.** *For the optimization problem of minimizing the $\mathcal{L}_{\text{CMDPO}}$ objective, defined as,*

$$
\mathcal{L}_{\text{CMDPO}}(\theta) = \mathbb{E}_{x,y\sim\pi_\theta}\left[((r(x,y) - \mathbb{E}_y r(x,y)) - (R_\theta(x,y) - \mathbb{E}_y R_\theta(x,y)))^2\right]
$$

*the policy $\pi_\theta = \pi^* = \pi_{ref}(y \mid x)\exp(r^*(x,y)/\beta)/Z$ is the unique optimal solution.*

**Theorem 2.** *An unbiased and consistent estimator of $\mathcal{L}_{\text{CMDPO}}(\theta)$ is given by*

$$
\frac{1}{N}\sum_{x,y\sim\pi_\theta}\frac{1}{k-1}\sum_{i=1}^{k}\left[(R_\theta(x,y_i) - \overline{R_\theta(x,y_i)}) - (r(x,y_i) - \overline{r(x,y_i)})\right]^2
$$

## 3.3 OPTIONAL STABILIZATION HEURISTICS

The CMDPO objective forms the core of our method. In practical training scenarios reward signals often contain heterogeneous components and generated responses may include redundant token segments. We introduce two optional enhancements to address these issues. These enhancements are general in nature and can be incorporated into other policy optimization methods.

### 3.3.1 DYNAMIC REWARD WEIGHTING STRATEGY

The rule-based reward mechanism is the true reward, typically divided into result and format rewards. During training, these two types of rewards differ in learning difficulty: format rewards tend to converge faster than result rewards. As such, more emphasis should be placed on the result reward rather than treating both equally.

Dynamic weights are proposed to modulate the model's sensitivity to different reward types. The combined reward is defined as:

$$
r = (2 - \alpha)r_{\text{result}} + \alpha r_{\text{format}}
\tag{7}
$$

Here, $\alpha$ is a monotonically decreasing function of $r_{\text{format}}$, given by:

$$
\alpha(r_{\text{format}}) = 1 - \sigma\left(s \cdot \left(\frac{1}{k}\sum_{i=1}^{k} r_{\text{format}} - \tau\right)\right)
\tag{8}
$$

where $s$ controls the steepness of the transition, $\sigma(\cdot)$ is the sigmoid function, and $\tau$ is a predefined baseline score.

### 3.3.2 TOKEN-LEVEL DISCRIMINATIVE LEARNING

In addition to loss function improvements, we consider token-level contributions. The core idea of outcome-based group optimization methods such as GRPO can be illustrated through an analogy: just as a student writes ten essays and a teacher determines which are good and which are bad, the model learns to produce better outputs by reinforcing the good ones and discarding the bad. However, what exactly makes an essay "good"? In reality, multiple essays written by the same author often contain redundant or overlapping content, which does not fundamentally determine the quality of the writing. Therefore, in learning from good and bad examples, we aim to reduce the influence of such non-discriminative components and focus more on the key features that truly distinguish high-quality outputs.

Table 1: Zero-shot pass@1 performance. Dashes (–) denote unavailable official scores.

| Methods | Avg | AIME24 | MATH-500 | AMC23 | Minerva | OlympiadBench |
|---|---|---|---|---|---|---|
| Llama-3.1-70B-Instruct | 35.7 | 16.7 | 64.6 | 30.1 | 35.3 | 31.9 |
| rStar-Math-7B | 26.7 | 78.4 | 47.5 | - | 47.1 | - |
| Eurus-2-7B-PRIME | 26.7 | 79.2 | 57.8 | 38.6 | 42.1 | 48.9 |
| DeepSeek-R1-Distill-Qwen-1.5B | 48.9 | 28.8 | 82.8 | 62.9 | 26.5 | 43.3 |
| Still-3-1.5B-Preview | 51.6 | 32.5 | 84.4 | 66.7 | 29.0 | 45.4 |
| Open-RS1 | 53.1 | 33.3 | 83.8 | 67.5 | 29.8 | 50.9 |
| GPG-RS1 | 55.7 | 33.3 | 87.6 | 77.5 | 29.4 | 50.5 |
| MDPO-RS1 | 52.6 | 26.7 | 84.4 | 75.0 | 26.8 | 50.2 |
| CMDPO-RS1 | 57.0 | 36.7 | 88.0 | 77.5 | 29.4 | 53.3 |
| Open-RS3 | 50.9 | 26.7 | 85.4 | 70.0 | 27.9 | 50.2 |
| GPG-RS3 | 55.5 | 33.3 | 85.0 | 80.0 | 26.8 | 52.4 |
| MDPO-RS3 | 53.8 | 33.3 | 83.8 | 75.0 | 26.5 | 50.2 |
| CMDPO-RS3 | 57.1 | 36.7 | 85.4 | 80.0 | 29.0 | 54.2 |

First, the general calculation for $\log \pi_\theta(y_i|x)$ is presented.

$$\log \pi_\theta(y_i \mid x) = \frac{1}{|y_i|} \sum_{t=1}^{|y_i|} \log \pi_\theta(y_{i,t} \mid x, y_{i,<t}) \qquad (9)$$

For a set of responses $y_i$ corresponding to a given input $x$, content that appears in all $y_i$ is considered as shared segments $\mathcal{Y}_o$. We reduce the weight of these shared segments using the following formulation:

$$\log \pi_\theta(y_i \mid x) = \frac{1}{M} \left[ \delta \sum_{y_{i,t} \in \mathcal{Y}_o} \log \pi_\theta(y_{i,t} \mid x, y_{i,<t}) + \sum_{y_{i,t} \notin \mathcal{Y}_o} \log \pi_\theta(y_{i,t} \mid x, y_{i,<t}) \right] \qquad (10)$$

where $M = |y_i| - |y_o| + \delta|y_o|$ is a normalizing factor and $\delta \in [0,1]$ denotes the corresponding weight coefficient.

## 4 EXPERIMENTS

### 4.1 EXPERIMENTAL SETUP

**Dataset and Benchmarks.** For unimodal, the training data includes the open-rs Dang & Ngo (2025) datasets. These datasets encompass a diverse range of problem types and difficulty levels. To evaluate the model's reasoning capabilities, we utilize five distinct math-focused benchmark datasets: AIME24, MATH-500 Hendrycks et al. (2021); Lightman et al. (2023), AMC23, Minerva Lewkowycz et al. (2022), and OlympiadBench Huang et al. (2024).

For multimodal, we address classification and reasoning grounding tasks following the Visual-RFT Liu et al. (2025d). We conducted a few-shot classification training on Flower102 Nilsback & Zisserman (2008), Pets37 Parkhi et al. (2012), FGVC Maji et al. (2013), and Cars196 Krause et al. (2013). In addition, we conducted training on 239 samples from the LISA Lai et al. (2024a). All evaluations use the corresponding test sets associated with each training dataset.

**Implementation Details.** All experiments were conducted on NVIDIA RTX 4090 24G GPUs. The Qwen2-VL-2B Wang et al. (2024b) is the backbone for multimodal tasks. The Still-3-1.5B-Preview Team et al. (2025b) is the backbone for unimodal tasks. We strictly followed the original codebase for each experiment to ensure consistent training and evaluation procedures. The parameter $\beta$ was set to 0.04. The parameter $s$ was set to 10 in the dynamic reward weight, and $\tau$ was set to 0.9. The weight of shared segments $\delta$ was set to 0.3.

### 4.2 COMPARATIVE RESULTS

**Mathematical Reasoning.** The Still-3-1.5B-Preview is used as the backbone to apply various RL under two settings (RS1 and RS3), including GRPO (Open) Dang & Ngo (2025), GPGChu et al.

Table 2: 4-shot results for fine-grained classification.

| Methods | Avg | Flower102 | Pets37 | FGVC | Cars196 |
|---|---|---|---|---|---|
| Qwen2-VL-2B | 56.0 | 54.8 | 66.4 | 45.9 | 56.8 |
| + SFT | 55.6 | 58.5 | 55.5 | 67.9 | 40.5 |
| + GRPO | 81.9 | 71.4 | 86.1 | 74.8 | 95.3 |
| + GPG | 86.0 | 73.0 | 87.1 | 86.8 | 97.1 |
| + MDPO | 84.3 | 74.4 | 86.8 | 78.8 | 97.3 |
| + CMDPO | 88.3 | 77.1 | 88.0 | 90.8 | 97.3 |

Table 3: Reasoning grounding on LISA.

| Methods | mIoU$_{test}$ | mIoU$_{val}$ | gIoU$_{test}$ |
|---|---|---|---|
| Qwen2-VL-2B | 26.9 | 30.1 | 25.3 |
| + SFT | 28.3 | 29.7 | 25.3 |
| + GRPO | 37.6 | 34.4 | 34.4 |
| + GPG | 51.5 | 53.4 | 49.5 |
| + MDPO | 51.8 | 51.3 | 49.6 |
| + CMDPO | 51.8 | 53.1 | 50.4 |

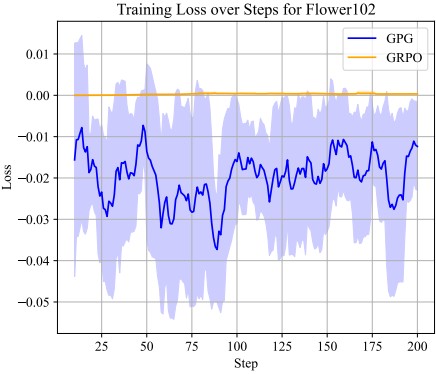
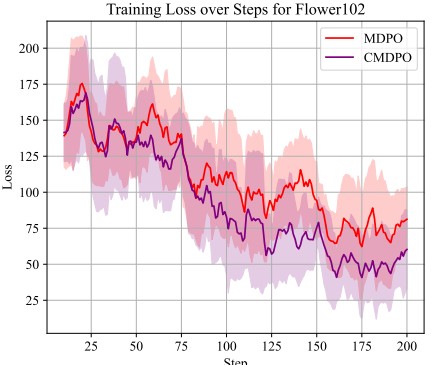

Figure 2: Loss curves on Flower102.

(2025), MDPOTeam et al. (2025a), and CMDPO. Several other baseline models are also included for comparison, such as Llama-3.1-70B-Instruct Meta (2024), rStar-Math-7B Guan et al. (2025), Eurus-2-7B-PRIME Cui et al. (2025), and DeepSeek-R1-Distill-Qwen-1.5B Team et al. (2025b).

As shown in Table 1, CMDPO achieves the highest performance among RL methods, with CMDPO-RS3 attaining an average accuracy of 57.1%, representing a +5.5% absolute gain over its base model, Still-3-1.5B-Preview (51.6%). Compared to other RL methods, CMDPO demonstrates superior improvements on challenging benchmarks. For instance, it achieves a +10.4% improvement on OlympiadBench (from 45.4% to 54.2%) and a +7.5% gain on AIME24, outperforming GPG and MDPO significantly. Notably, CMDPO-RS3 achieves an average improvement of +1.6% over GPG-RS3, with even larger gains observed on AMC23 (+2.5%) and OlympiadBench (+1.8%).

**Classification.** Table 2 shows the results of 4-shot fine-grained image classification. The baseline model, Qwen2-VL-2B, achieves an average accuracy of 56.0%, with a slight drop to 55.6% after supervised fine-tuning (SFT). Reinforcement learning methods lead to significant performance gains. CMDPO outperforms all other methods on all datasets, reaching an average accuracy of 88.3%. It improves over MDPO by approximately 4% on average and by 12% on FGVC.

**Reasoning Grounding.** As shown in Table 3, CMDPO achieves the best gIoU of 50.4%, improving over Qwen2-VL-2B by +25.1 %, nearly doubling the performance. Its mIoU$_{test}$ also increases by +24.9 % (from 26.9% to 51.8%). Compared to MDPO and GPG, CMDPO maintains comparable mIoU while slightly improving gIoU$_{test}$, indicating better boundary alignment and generalization.

### 4.3 LOSS CONVERGENCE COMPARISON

To investigate the convergence advantages of CMDPO, Figure 2 visualizes the loss curves during training on the Flower102 dataset. It can be observed that GRPO exhibits only minor variations in loss values, while GPG shows fluctuating loss curves without a clear convergence point. In contrast, both MDPO and CMDPO demonstrate a clear trend of loss reduction and convergence, with CMDPO showing a more significant decrease in loss than MDPO. Loss curves for other datasets are provided in Appendix A.12.

Table 4: Impact of shared segments weight $\delta$.

| $\delta$ | Avg | Flower102 | Pets37 | FGVC | Cars196 |
|---|---|---|---|---|---|
| 0.1 | 87.0 | 74.7 | 87.2 | 89.0 | 96.9 |
| 0.3 | 88.3 | 77.1 | 88.0 | 90.8 | 97.3 |
| 0.5 | 87.9 | 76.3 | 88.0 | 90.0 | 97.2 |
| 0.7 | 88.6 | 77.5 | 88.3 | 91.0 | 97.4 |
| 1 | 87.5 | 75.3 | 87.8 | 89.9 | 97.1 |

Table 5: Impact of KL divergence coefficient.

| $\beta$ | Avg | Flower102 | Pets37 | FGVC | Cars196 |
|---|---|---|---|---|---|
| 0.001 | 88.31 | 77.10 | 88.03 | 90.79 | 97.30 |
| 0.01 | 87.62 | 74.30 | 88.06 | 90.73 | 97.40 |
| 0.04 | 87.40 | 73.37 | 88.33 | 90.61 | 97.30 |
| 0.1 | 86.42 | 73.73 | 85.31 | 89.89 | 96.77 |

Table 6: Analysis of format and result reward weights.

| $\alpha$ | Avg | Flower102 | Pets37 | FGVC | Cars196 |
|---|---|---|---|---|---|
| 0 | 86.3 | 74.2 | 86.1 | 88.5 | 96.3 |
| 0.25 | 87.1 | 75.1 | 86.6 | 89.9 | 96.9 |
| 0.5 | 86.9 | 75.1 | 87.1 | 88.9 | 96.5 |
| 0.75 | 87.6 | 75.8 | 87.5 | 89.5 | 97.4 |
| 1 | 87.8 | 76.7 | 87.5 | 90.0 | 97.1 |
| DRW | 88.3 | 77.1 | 88.0 | 90.8 | 97.3 |



Figure 3: Generalization performance.

## 4.4 ABLATION STUDY

**Ablation on $\delta$.** We further evaluated the sensitivity of our method to the hyperparameter $\delta$, which controls the weight of the shared segments. As shown in Table 4, the model achieves the best performance when $\delta = 0.7$. Increasing $\delta$ places excessive emphasis on shared tokens, affecting the model's ability to learn discriminative differences among diverse solutions.

**Ablation on $\beta$.** Table 5 reports the experimental results under different values of $\beta$, which controls the strength of the KL divergence penalty. A larger $\beta$ imposes a stronger regularization, encouraging the model to stay closer to the previous policy during training. We observe that setting $\beta = 0.001$ yields the best performance across all datasets. As $\beta$ increases, the penalty becomes more restrictive, reducing the model's tolerance for deviation from the prior policy, leading to a drop in performance.

**Ablation on $\alpha$.** Table 6 presents the performance under different values of $\alpha$, which controls the weight of the format reward. The format reward encourages the model to explore intermediate reasoning steps. The model performs the worst when the format reward is removed (i.e., $alpha = 0$). However, an overly dominant format reward can saturate early in training, diminishing the model's focus on the result reward. Although lowering $\alpha$ can help shift attention toward result quality, a fixed value of $\alpha$ may not generalize well across diverse samples. In contrast, the proposed dynamic reward weighting (DRW) consistently outperforms fixed $\alpha$ settings across all datasets. DRW adaptively relaxes format constraints during training (see Appendix A.13), enabling the model to explore more reasoning paths that lead to correct answers.

## 4.5 IMPACT OF GROUP NUMBER

Table 7 presents the performance of MDPO and CMDPO under varying group numbers $k$. Overall, CMDPO consistently outperforms MDPO across all settings and datasets, demonstrating its robustness and effectiveness. When $k = 2$, CMDPO shows substantial gains over MDPO, with improvements of +7.9 on Flower102, +5.0 on Pets37, and +4.6 on FGVC. The results highlight CMDPO's ability to maintain performance even when the sample size per group is limited.

As $k$ increases, both methods benefit from more stable reward statistics, but CMDPO maintains a consistent lead. At $k = 16$, CMDPO achieves the highest average accuracy of 90.4, compared to 87.0 for MDPO. These results confirm that the group centering strategy effectively eliminates partition function bias in MDPO.

Table 7: Impact of group number $k$ on the performance of MDPO and CMDPO. When $k$ is relatively small, the performance degradation of MDPO is greater than that of CMDPO.

| $k$ | Average | | Flower102 | | Pets37 | | FGVC | | Cars196 | |
|---|---|---|---|---|---|---|---|---|---|---|
| | MDPO | CMDPO | MDPO | CMDPO | MDPO | CMDPO | MDPO | CMDPO | MDPO | CMDPO |
| 2 | 78.5 | 83.4 | 66.7 | 74.6 | 81.7 | 86.7 | 75.3 | 79.9 | 90.4 | 92.3 |
| 4 | 80.9 | 85.9 | 71.4 | 75.1 | 82.2 | 87.4 | 77.2 | 85.9 | 92.9 | 95.2 |
| 8 | 84.3 | 88.3 | 74.4 | 77.1 | 86.8 | 88.0 | 78.8 | 90.8 | 97.3 | 97.3 |
| 16 | 87.0 | 90.4 | 76.3 | 79.3 | 87.6 | 92.8 | 86.5 | 91.9 | 97.6 | 97.4 |

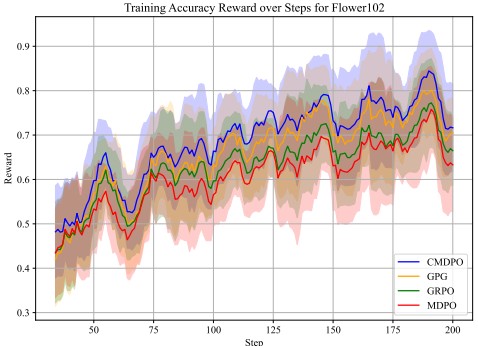 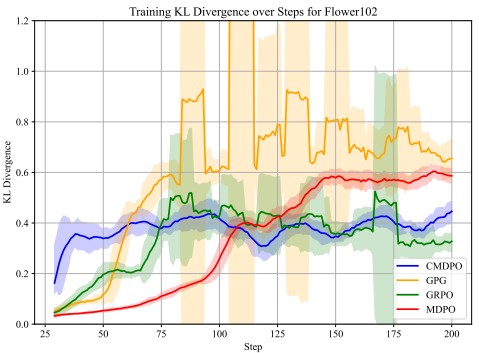

Figure 4: Comparison of smoothed training result reward and KL divergence on Flower102.

## 4.6 GENERALIZATION ANALYSIS

To evaluate the generalization capabilities of various methods, we trained models on one domain and tested their average performance across other domains. The results are shown in Figure 3. The results indicate that GPG has the poorest generalization performance, likely due to a lack of KL divergence constraints. The CMDPO demonstrates significantly better generalization capabilities than other methods. A detailed generalization analysis is provided in Appendix A.11.

## 4.7 REWARD AND KL DIVERGENCE ANALYSIS

Figure 4 plots the result reward and KL divergence curves during training on Flower102 to better compare different RL methods. The result reward trends are generally consistent across methods. Differences among these methods, as detailed in Appendix A.5, primarily involve KL divergence constraints and clipping strategies, which do not significantly impact reward changes. CMDPO shows superior result reward improvement with a smaller increase in KL divergence, indicating that CMDPO achieves higher rewards while staying close to the reference model, demonstrating a higher KL-reward conversion efficiency. The reward and KL divergence curves on other datasets are provided in Appendix A.13.

## 5 CONCLUSION

This study proposes CMDPO, a theoretically grounded reinforcement learning framework that eliminates the need to estimate the intractable partition function $\log Z$ via a group centering strategy. This design preserves the stability and convergence properties of MDPO while improving computational efficiency. To further enhance learning quality, we introduced dynamic reward weighting strategy for balancing result and format rewards, and token-level discriminative learning for finer control over generation. Extensive experiments demonstrate that CMDPO consistently outperforms existing approaches in performance, generalization, and interpretability, offering a robust and scalable solution.

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

# A APPENDIX

## A.1 ESTIMATION OF $\log Z$ WITH AVERAGE REWARD

We draw $k$ responses $\{y_i\}_{i=1}^k$ from the policy $\pi_{\theta_i}(y|x)$, and approximate the expectation as follows:

$$\beta \log Z(x) = \beta \sum_y \pi_{\theta_i}(y \mid x) \exp(\frac{r(x,y)}{\beta})$$

$$\approx \beta \log \left[ \frac{1}{k} \sum_{i=1}^k \exp \left( \frac{r(x,y_i)}{\beta} \right) \right] \tag{11}$$

It is important to note that the distribution $\pi_{\theta_i}(y|x)$ is not explicitly canceled out in this approximation; rather, it is implicitly represented through the sampling process.

Under assumptions for sufficiently large $k$, we can further approximate the sample mean of exponentials using Jensen's inequality or Laplace method-like arguments:

$$\beta \log Z(x) \approx \beta \log \left[ \frac{1}{k} \sum_{i=1}^k \exp \left( \frac{r(x,y_i)}{\beta} \right) \right]$$

$$\approx \beta \log \left[ \frac{1}{k} \cdot k \cdot \exp \left( \frac{\overline{r(x,\{y_i\})}}{\beta} \right) \right]$$

$$\approx \overline{r(x,\{y_i\})} \tag{12}$$

This approximation avoids the need for normalization over the entire output space. However, Equations (11) and (12) rely on a sufficiently large group number $k$ to ensure approximation accuracy. In most practical scenarios, the group number $k$ is typically limited to fewer than 8, which can lead to significant approximation bias.

## A.2 EXISTENCE OF A UNIQUE OPTIMAL SOLUTION

The proposed CMDPO loss admits a unique optimal solution, which coincides with the optimal policy for maximizing reward under a KL-divergence constraint, which ensures the optimization problem is well-posed and has a single global minimum.

**Theorem 1.** *For the optimization problem of minimizing the $\mathcal{L}_{CMDPO}$ objective, defined as,*

$$\mathcal{L}_{CMDPO}(\theta) = \mathbb{E}_{x,y\sim\pi_\theta} \left[ ((r(x,y) - \mathbb{E}_y r(x,y)) - (R_\theta(x,y) - \mathbb{E}_y R_\theta(x,y)))^2 \right]$$

*the policy $\pi_\theta = \pi^* = \pi_{ref}(y \mid x) \exp(r^*(x,y)/\beta)/Z$ is the unique optimal solution.*

*Proof.* We proceed by showing sufficiency and necessity.

**Sufficiency:** If the policy is optimal solution, then $\pi_\theta(y|x) = \pi^*(y|x)$.

Assume that there exists an optimal policy $\pi_\theta(y|x) \neq \pi^*(y|x)$. Then, we have $r(x,y) - \mathbb{E}_y r(x,y) = R(x,y) - \mathbb{E}_y R(x,y)$.

According to the variant of the closed-form solution, we can rewrite $r(x,y)$ and $R_\theta(x,y)$ as follows:

$$\beta \log \pi^*(y|x) - \mathbb{E}_y r(x,y) = \beta \log \pi_\theta(y|x) - \mathbb{E}_y R(x,y)$$

Rearranging the above expression yields:

$$\pi_\theta(y \mid x) = \exp \left( \frac{\mathbb{E}_y R(x,y) - \mathbb{E}_y r(x,y)}{\beta} \right) \pi^*(y \mid x) \tag{13}$$

If the policies are valid, then it must hold that $\pi_\theta(y|x), \pi^*(y|x) > 0$ for any $(x, y)$. And they satisfy the normalization constraint $\sum_y \pi_\theta(y \mid x) = \sum_y \pi^*(y \mid x) = 1$. Therefore, the following Equation

(14) holds:

$$\sum_y \pi_\theta(y \mid x) = \exp\left(\frac{\mathbb{E}_y R(x,y) - \mathbb{E}_y r(x,y)}{\beta}\right) \sum_y \pi^*(y \mid x)$$

$$\implies \quad \exp\left(\frac{\mathbb{E}_y R(x,y) - \mathbb{E}_y r(x,y)}{\beta}\right) = 1 \tag{14}$$

Substituting Equation (14) into Equation (13) yields $\pi_\theta(y|x) = \pi^*(y|x)$, which contradicts the assumption that $\pi_\theta(y|x) \neq \pi^*(y|x)$.

**Necessity:** If $\pi_\theta(y|x) = \pi^*(y|x)$, then the policy is optimal solution.

Since $\mathcal{L}_{\text{CMDPO}}$ employs a squared loss function, the loss is always non-negative, i.e., $\mathcal{L}_{\text{CMDPO}} \geq 0$. When $\pi_\theta = \pi^*$, the implicit reward $R_\theta(x,y)$ coincides exactly with the true reward $r(x,y)$, resulting in $\mathcal{L}_{\text{CMDPO}}$ achieving its theoretical minimum value of zero.

Since necessity and sufficiency hold, the optimal policy is uniquely $\pi^*$. $\qquad\square$

A unique global optimum guarantees that minimizing $\mathcal{L}_{\text{CMDPO}}$ leads to the optimal policy that maximizes expected reward while remaining close to the reference policy regarding behavior distribution, which ensures a well-behaved optimization landscape, free from suboptimal local minima, and guarantees convergence to a single, interpretable policy. This policy achieves the best possible trade-off between reward maximization and behavioral consistency concerning the reference.

$$\nabla_\theta \mathcal{L}_{\text{CMDPO}}(\theta) = \frac{1}{2\beta} \frac{1}{N} \nabla_\theta \sum_{x,y \sim \pi_\theta} \frac{1}{k} \sum_{i=1}^k \left[ \left( r(x,y_i) - \overline{r(x,\{y_i\})} - \left( R_\theta(x,y_i) - \overline{R_\theta(x,\{y_i\})} \right) \right)^2 \right] \tag{15}$$

$$= -\frac{1}{kN} \sum_{x,y \sim \pi_\theta} \sum_{i=1}^k \left[ r(x,y_i) - \overline{r(x,\{y_i\})} - \left( R_\theta(x,y_i) - \overline{R_\theta(x,\{y_i\})} \right) \right]$$

$$\nabla_\theta \left( R_\theta(x,y_i) \right) - \overline{R_\theta(x,\{y_i\})} \right) \tag{16}$$

$$= -\frac{1}{kN} \sum_{x,y \sim \pi_\theta} \sum_{i=1}^k \left[ r(x,y_i) - \overline{r(x,\{y_i\})} - \left( R_\theta(x,y_i) - \overline{R_\theta(x,\{y_i\})} \right) \right]$$

$$\nabla_\theta \left( \log \frac{\pi_\theta(y_i \mid x)}{\pi_{\text{ref}}(y_i \mid x)} - \overline{\log \frac{\pi_\theta(\{y_i\} \mid x)}{\pi_{\text{ref}}(\{y_i\} \mid x)}} \right) \tag{17}$$

$$= -\frac{1}{kN} \sum_{x,y \sim \pi_\theta} \sum_{i=1}^k \left[ r(x,y_i) - \overline{r(x,\{y_i\})} - \left( R_\theta(x,y_i) - \overline{R_\theta(x,\{y_i\})} \right) \right] \nabla_\theta \log \frac{\pi_\theta(y_i \mid x)}{\pi_{\text{ref}}(y_i \mid x)} \tag{18}$$

$$= -\frac{1}{kN} \sum_{x,y \sim \pi_\theta} \sum_{i=1}^k \left[ r(x,y_i) - \overline{r(x,\{y_i\})} - \left( R_\theta(x,y_i) - \overline{R_\theta(x,\{y_i\})} \right) \right] \nabla_\theta \log \pi_\theta(y_i \mid x) \tag{19}$$

## A.3 Unbiasedness and Consistency of the Estimator

The Monte Carlo estimator of the $\mathcal{L}_{\text{CMDPO}}(\theta)$ is both unbiased and consistent.

**Theorem 2.** *An unbiased and consistent estimator of $\mathcal{L}_{CMDPO}(\theta)$ is given by*

$$\frac{1}{N} \sum_{x,y \sim \pi_\theta} \frac{1}{k-1} \sum_{i=1}^k \left[ (R_\theta(x,y_i) - \overline{R_\theta(x,y_i)}) - (r(x,y_i) - \overline{r(x,y_i)}) \right]^2$$

*Proof.* We refer to the expressions for variance and covariance given in Equation (6) to proceed.

$$\mathcal{L}_{\text{CMDPO}}(\theta) = \mathbb{E}_{x,y\sim\pi_\theta}[(r(x,y) - \mathbb{E}_y r(x,y))^2 + \underbrace{(R_\theta(x,y) - \mathbb{E}_y R_\theta(x,y))^2}_{\text{Variance}}$$

$$- 2\underbrace{(R_\theta(x,y) - \mathbb{E}_y R_\theta(x,y))(r(x,y) - \mathbb{E}_y r(x,y))}_{\text{Covariance}}]$$

Since sample variance and sample covariance are unbiased and consistent estimators of variance and covariance, the theorem has been proven. $\square$

Unbiasedness ensures that the estimator's expected value equals the estimated quantity's true value, which is crucial during training, as it prevents systematic errors from accumulating over iterations, which could otherwise lead the model astray. Consistency guarantees that the estimator converges in probability to the true parameter value as the number of samples increases. Consistency implies that with sufficient data, we can achieve arbitrarily accurate estimates of the loss function, enabling stable and reliable optimization. Therefore, the estimator supports reliable generalization and enables stable optimization in practical learning settings.

### A.4 CONNECTION TO POLICY GRADIENTS

By taking the derivative of $\mathcal{L}_{\text{CMDPO}}$ with respect to $\theta$, we obtain the Equations (15) – (19).

Here, $N$ denotes the total number of samples. Notably, since the mean of the implicit reward and $\log \pi_{\text{ref}}$ are constant terms, and the coefficients satisfy the zero-sum property (as discussed in Section 3), we can derive Equation (19) from Equation (17). This expression reveals a close connection between CMDPO and the standard policy gradient. Now, the general form of the policy gradient is presented.

$$\nabla_\theta J(\theta) = \mathbb{E}_{x,y\sim\pi_\theta}[r(x,y)\nabla_\theta \log \pi_\theta(y\mid x)] = \frac{1}{kN}\sum_{x,y_i\sim\pi_\theta}\sum_{i=1}^{k} r(x,y_i)\nabla_\theta \log \pi_\theta(y_i\mid x) \quad (20)$$

This expression closely resembles the gradient structure of our loss function. To align it with the gradient descent formulation used in CMDPO, we can introduce a negative sign to convert the maximization of the expected reward into a minimization problem. Moreover, apart from the true reward $r(x,y_i)$, the remaining part of the coefficient can be interpreted as a baseline term that helps reduce variance in the gradient estimate. Here, we present the policy gradient formulation with such a baseline.

$$\nabla_\theta J(\theta) = \frac{1}{kN}\sum_{x,y_i\sim\pi_\theta}\sum_{i=1}^{k}[r(x,y_i) - b]\nabla_\theta \log \pi_\theta(y_i\mid x) \quad (21)$$

In addition, the CMDPO loss offers several unique advantages:

- **Centered Rewards:** Unlike the raw rewards used in standard policy gradients, we employ centered rewards, which help reduce variance and lead to more stable learning signals across different samples.

- **Implicit Reward as Learning Signal:** The gradient of our loss function depends on the difference between the true reward and the implicit reward, rather than just the true reward itself, encouraging the model to maximize the reward and align its internal reward estimation with the ground truth.

In summary, CMDPO can be viewed as an extension of policy gradients, enhanced by reward centering and implicit reward modeling. These improvements provide the model with a more robust and stable update direction.

## A.5 DISCUSSION ON KL

By computing the derivative of Equation (6) concerning $\theta$, an alternative form of $\nabla_\theta \mathcal{L}_{\text{CMDPO}}(\theta)$ can be obtained.

$$\nabla_\theta \mathcal{L}_{\text{CMDPO}}(\theta) = -2\mathbb{E}_{x,y\sim\pi_\theta}\big[(r(x,y) - \mathbb{E}_y r(x,y))$$

$$\nabla_\theta \log \pi_\theta(y \mid x) - \frac{1}{2}\nabla_\theta(R_\theta(x,y) - \mathbb{E}_y R_\theta(x,y))^2\big]$$

Here, the first term is the reward-normalized policy gradient, similar to methods like GRPO. The second term can be viewed as a KL constraint that prevents the policy from deviating too far from $\pi_{\text{ref}}$, which is the main difference from MDPO in terms of gradient formulatio[1].

$$\nabla_\theta \mathcal{L}_{\text{MDPO}}(\theta) = -2\mathbb{E}_{x,y\sim\pi_\theta}\left[(r(x,y) - \mathbb{E}_y r(x,y))\nabla_\theta \log \pi_\theta(y \mid x) - \frac{\beta}{2}\nabla_\theta\left(\log\frac{\pi_\theta(y \mid x)}{\pi_{\text{ref}}(y \mid x)}\right)^2\right]$$

The gradient of MDPO is similar to that of GRPO, except that the estimation of the KL divergence is changed from $k_3$ to $k_2$. Our CMDPO, on the other hand, introduces a normalized variant of $k_2$, denoted as $k_2^{\text{norm}}$, in the gradient formulation. For clarity in subsequent discussions, we focus only on the gradient component corresponding to the KL divergence term. We next present the formal definitions of $k_1$, $k_2$, $k_3$, and $k_2^{\text{norm}}$.

$$k_1 = \log\frac{\pi_\theta(y \mid x)}{\pi_{\text{ref}}(y \mid x)} \tag{22}$$

$$k_2 = \frac{1}{2}\left(\log\frac{\pi_\theta(y \mid x)}{\pi_{\text{ref}}(y \mid x)}\right)^2 \tag{23}$$

$$k_2^{\text{norm}} = \frac{1}{2}\left(\log\frac{\pi_\theta(y \mid x)}{\pi_{\text{ref}}(y \mid x)} - \mathbb{E}_y \log\frac{\pi_\theta(y \mid x)}{\pi_{\text{ref}}(y \mid x)}\right)^2 \tag{24}$$

$$k_3 = \frac{\pi_{\text{ref}}(y \mid x)}{\pi_\theta(y \mid x)} - 1 - \log\frac{\pi_{\text{ref}}(y \mid x)}{\pi_\theta(y \mid x)} \tag{25}$$

Existing KL constraint methods can be broadly categorized into two types: one treats the KL divergence as a penalty term in the reward, and the other incorporates it directly into the loss function. Different constraint formulations impose different requirements on how the KL divergence is estimated. We first consider the latter approach that uses KL as part of the loss andz relevant to $\mathcal{L}_{\text{CMDPO}}$. In this setting, the KL divergence must remain non-negative (as $k_1$ may take negative values), otherwise a "short-circuit" behavior could emerge: the policy $\pi_\theta$ might continuously reduce the KL value toward increasingly negative values to satisfy the optimization objective, thereby ignoring the true policy gradient. Next, we analyze the suitability of the four KL divergence estimators ($k_1$, $k_2$, $k_3$, $k_2^{\text{norm}}$) from a gradient-based perspective.

$$\nabla_\theta k_1 = \nabla_\theta \log \pi_\theta(y \mid x) \tag{26}$$

$$\nabla_\theta k_2 = \log\frac{\pi_\theta(y \mid x)}{\pi_{\text{ref}}(y \mid x)} \cdot \nabla_\theta \log \pi_\theta(y \mid x) \tag{27}$$

$$\nabla_\theta k_2^{\text{norm}} = (\log\frac{\pi_\theta(y \mid x)}{\pi_{\text{ref}}(y \mid x)} - \mathbb{E}_y \log\frac{\pi_\theta(y \mid x)}{\pi_{\text{ref}}(y \mid x)}) \cdot \nabla_\theta \log \pi_\theta(y \mid x) \tag{28}$$

$$\nabla_\theta k_3 = -\left(\frac{\pi_{\text{ref}}(y\mid x)}{\pi_\theta(y\mid x)} - 1\right) \cdot \nabla_\theta \log \pi_\theta(y \mid x) \tag{29}$$

It can be observed that the gradient of $k_1$ is always negative, which leads to a continuous decrease in the policy model's probability—a result consistent with our earlier analysis. In contrast, the gradient directions of $k_2$ and $k_3$ are determined by the relative magnitudes of $\pi_\theta$ and $\pi_{\text{ref}}$. Notably, $k_2$ exhibits symmetry between $\pi_\theta$ and $\pi_{\text{ref}}$, whereas $k_3$ does not possess this property. The proposed $k_2^{\text{norm}}$ shares similarities with $k_2$, but incorporates a mean normalization term, improving its stability.

Furthermore, $k_2^{\text{norm}}$ possesses additional practical interpretations, as formalized in the following theorem:

---

[1] It should be noted that the MDPO gradient here does not consider iterative training and instead uses $\pi_{\text{ref}}$ in place of $\pi_{\theta_i}$.

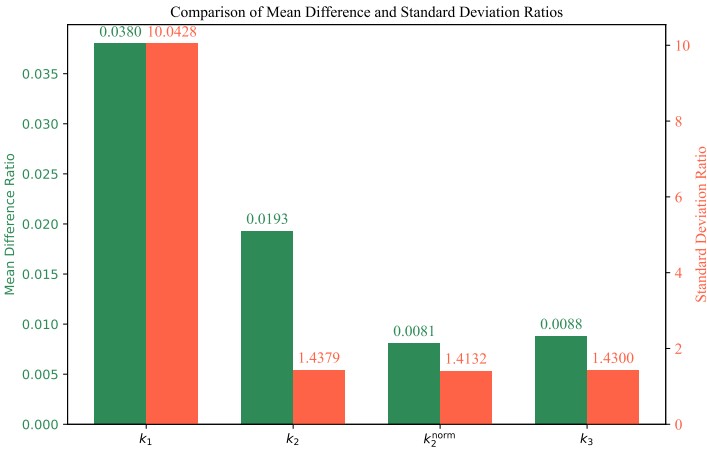

Figure 5: Comparison of different KL divergence estimates.

**Theorem 3.** $\forall \pi_\theta > 0, \pi_{ref} > 0$, *minimizing $k_2^{norm}$ is equivalent to maximizing the covariance between $\log \pi_{ref}$ and $\log \pi_\theta$, while simultaneously minimizing the variance of $\log \pi_\theta$, i.e.,*

$$\min_{\pi_\theta} k_2^{norm} \Leftrightarrow \max_{\pi_\theta} \mathrm{Cov}\left(\log \pi_\theta, \log \pi_{\text{ref}}\right) - \mathrm{Var}\left(\log \pi_\theta\right)$$

*Proof.* Expanding the squared term in $k_2^{\text{norm}}$, we have,

$$k_2^{\text{norm}} = \frac{1}{2}\left(\log \frac{\pi_\theta}{\pi_{\text{ref}}} - \mathbb{E}\log \frac{\pi_\theta}{\pi_{\text{ref}}}\right)^2$$

$$= \frac{1}{2}[(\log \pi_\theta - \mathbb{E}\log \pi_\theta) - (\log \pi_{\text{ref}} - \mathbb{E}\log \pi_{\text{ref}})]^2$$

$$= \frac{1}{2}[(\log \pi_{\text{ref}} - \mathbb{E}\log \pi_{\text{ref}})^2 + (\log \pi_\theta - \mathbb{E}\log \pi_\theta)^2 - 2(\log \pi_{\text{ref}} - \mathbb{E}\log \pi_{\text{ref}})(\log \pi_\theta - \mathbb{E}\log \pi_\theta)]$$

$$= \frac{1}{2}[\mathrm{Var}(\log \pi_{\text{ref}}) + \mathrm{Var}(\log \pi_\theta) - 2\mathrm{Cov}\left(\log \pi_\theta, \log \pi_{\text{ref}}\right)]$$

where $\mathrm{Var}(\log \pi_{\text{ref}})$ is a constant. The proof is complete. $\square$

To more intuitively illustrate the differences among these estimates, Figure 5 visualizes the mean difference ratio and standard deviation ratio of various KL estimates compared to the true KL divergence. The formulas for calculating the mean difference ratio and standard deviation ratio are as follows:

$$\text{Mean Difference Ratio} = \frac{\mu(\hat{kl}) - kl}{kl}$$

$$\text{Standard Deviation Ratio} = \frac{\mathrm{std}(\hat{kl})}{kl}$$

Where $kl$ and $\hat{kl}$ respectively represent the true KL divergence and the estimator of KL divergence. The functions $\mu(\cdot)$ and $\mathrm{std}(\cdot)$ respectively represent the operations of calculating the mean and the standard deviation. The results indicate that $k_2^{\text{norm}}$ has lower bias and variance than other methods, performing slightly better than $k_3$.

For the KL constraint as a penalty in the reward, we will discuss it in the next section.

### A.6 EXTRA DISCUSSION ON KL

Derive the general form of the policy loss when KL divergence is used as a reward penalty.

$$J(\theta) = \mathbb{E}_{x,y}\left[(r(x,y) - \beta \mathrm{KL}(\pi_{\text{ref}}\|\pi_\theta))\log \pi_\theta(y|x)\right] \tag{30}$$

where the KL term is $-\beta \, \text{KL}(\pi_{\text{ref}} \| \pi_\theta) \log \pi_\theta(y \mid x)$.

When KL divergence is used as a reward penalty, it can be categorized into two types depending on whether the KL term participates in gradient computation. First, consider the case where the KL divergence does not participate in gradient computation — in this setting, the KL term acts as a coefficient in the policy gradient.

Since $k_2, k_2^{\text{norm}}$, and $k_3$ are always non-negative, the policy probability will continuously decrease under these penalties. Only $k1$, which can take negative values, is suitable for such scenarios. From the gradient forms of different KL estimators discussed in the previous section (Equations (27)–(29)), we can derive equivalent expressions for $k_2, k_2^{\text{norm}}$, and $k_3$ under this setup: $k_2' = \log \frac{\pi_\theta(y|x)}{\pi_{\text{ref}}(y|x)}, k_2'^{\text{norm}} = \log \frac{\pi_\theta(y|x)}{\pi_{\text{ref}}(y|x)} - \mathbb{E}_y \log \frac{\pi_\theta(y|x)}{\pi_{\text{ref}}(y|x)}, k_3' = -\left( \frac{\pi_{\text{ref}}(y|x)}{\pi_\theta(y|x)} - 1 \right)$. Notably, the estimate $k_2'$ is equivalent to $k_1$. Starting from $k_1$ (where KL is used as a reward penalty), one can also derive its equivalent form when KL is treated as a loss term. This derivation is omitted here for brevity.

When the KL divergence does participate in gradient computation, the gradients of the four estimates are given as follows:

$$\nabla_\theta k_1 = \beta \left( (\log \pi_{ref} - 2 \log \pi_\theta) \nabla_\theta \log \pi_\theta \right) \tag{31}$$

$$\nabla_\theta k_2 = -3 \left( \frac{\log \pi_\theta}{\log \pi_{ref}} \right)^2 \nabla_\theta log \pi_\theta \tag{32}$$

$$\nabla_\theta k_2^{\text{norm}} = -3 \left( \frac{\log \pi_\theta}{\log \pi_{ref}} - \mathbb{E} \frac{\log \pi_\theta}{\log \pi_{ref}} \right)^2 \nabla_\theta log \pi_\theta \tag{33}$$

$$\nabla_\theta k_3 = \left[ \left( \frac{\pi_{ref}}{2\pi_\theta} + 2 \right) \log(\pi_\theta) - \frac{\pi_{ref}}{\pi_\theta} + 1 - \log(\pi_{ref}) \right] \nabla_\theta log \pi_\theta$$

For $k_1$, the direction of the policy gradient still depends on both $\pi_\theta$ and $\pi_{\text{ref}}$. When $\pi_{\text{ref}} > 2\pi_\theta$, the gradient direction is positive, increasing policy probability. Conversely, the gradient direction is negative when $\pi_{\text{ref}} < 2\pi_\theta$. In both cases, the update drives policy distribution closer to the reference distribution.

For $k_2$ and $k_2^{\text{norm}}$, the gradient direction is always negative, resulting in a continuous decrease in policy probability.

For $k_3$, the behavior is more intricate. We define the function $f(\pi_\theta, \pi_{\text{ref}}) = \left( \frac{\pi_{\text{ref}}}{2\pi_\theta} + 2 \right) \log(\pi_\theta) - \frac{\pi_{\text{ref}}}{\pi_\theta} + 1 - \log(\pi_{\text{ref}})$ to analyze the gradient behavior.

When $\pi_\theta = \pi_{\text{ref}} \in (0, 1)$, we have $f(\pi_\theta, \pi_\theta) = \frac{3}{2} \log \pi_\theta < 0$, which implies that the gradient direction is negative.

When $\pi_\theta < \pi_{\text{ref}} \in (0, 1)$, we can rewrite $f(\pi_\theta, \pi_{\text{ref}})$ as:

$$f(\pi_\theta, \pi_{\text{ref}}) = \left( \frac{\pi_{\text{ref}}}{2\pi_\theta} + 1 \right) \log(\pi_\theta) + 1 - \frac{\pi_{\text{ref}}}{\pi_\theta} + \log \left( \frac{\pi_\theta}{\pi_{\text{ref}}} \right).$$

Each term in this expression is negative:

- $\left( \frac{\pi_{\text{ref}}}{2\pi_\theta} + 1 \right) \log(\pi_\theta) < 0$,
- $1 - \frac{\pi_{\text{ref}}}{\pi_\theta} < 0$,
- $\log \left( \frac{\pi_\theta}{\pi_{\text{ref}}} \right) < 0$.

Thus confirming that $f(\pi_\theta, \pi_{\text{ref}}) < 0$, and the gradient direction remains negative.

When $\pi_\theta > \pi_{\text{ref}} \in (0, 1)$, the sign of $f(\pi_\theta, \pi_{\text{ref}})$ is not constant — it may be either positive or negative depending on the specific values of $\pi_\theta$ and $\pi_{\text{ref}}$.

In summary, when $\pi_\theta < \pi_{\text{ref}}$, as well as when $\pi_\theta > \pi_{\text{ref}}$ and the gradient direction is negative, the policy distribution is driven away from the reference distribution, which is undesirable.

Table 8: CMDPO performance under different group normalization factors. F_norm = 1 uses fixed scaling, std uses standard deviation, clip_std = proposed clipped std.

| Methods | Avg | Flower102 | Pets37 | FGVC | Cars196 |
|---|---|---|---|---|---|
| Qwen2-VL-2B | 56.0 | 54.8 | 66.4 | 45.9 | 56.8 |
| + CMDPO (F_norm = 1) | 88.3 | 77.1 | 88.0 | 90.8 | 97.3 |
| + CMDPO (F_norm = std) | 86.5 | 75.4 | 87.3 | 86.8 | 96.5 |
| + CMDPO (F_norm = clip_std) | 88.9 | 78.0 | 89.2 | 91.4 | 97.1 |

## A.7 STANDARD DEVIATION IN ADVANTAGE

In Appendix A.5, we analyzed the gradient-based differences between CMDPO and GRPO, identifying two key distinctions: (1) the estimator for KL divergence; and (2) whether the advantage is normalized by its standard deviation. The former has been thoroughly discussed. This section focuses on the latter, namely, the role of standard deviation in advantage estimation.

First, we present the standard advantage calculation formula in GRPO,

$$\hat{A}(x, y) = \frac{r_{(x, y)} - \mu(r(x, y))}{\text{std}(r(x, y))}$$

The standardization acts as a scaling mechanism based on learning difficulty. The standard deviation is large when response scores vary widely, leading to smaller normalized advantages. In contrast, when scores are close, the standard deviation is small, which magnifies small differences, encouraging the model to pay attention to fine-grained distinctions.

Despite its benefits, standardizing advantages can introduce numerical instability. In extreme cases where all response scores are identical, the standard deviation becomes zero, causing the advantage to diverge. Even when scores are not exactly equal, near-zero variance—caused by quantization error or modeling bias—can lead to inflated advantage values. Such instability, accelerating convergence in GRPO, can harm generalization and result in oscillatory behavior. Therefore, standard deviation is not used for normalization in CMDPO.

One potential remedy is to apply a clipped standard deviation,

$$\text{clip\_std}(s) = \max(\min(s, \sigma_{\max}), \sigma_{\min}),$$

where $\sigma_{\min}$ and $\sigma_{\max}$ are predefined thresholds. The normalized advantage then becomes,

$$\hat{A}_{\text{clipped}}(x, y) = \frac{r(x, y) - \mu(r(x, y))}{\text{clip\_std}(\text{std}(r(x, y)))}. \tag{34}$$

The clipped standard deviation preserves the benefits of normalization while safeguarding against extreme scaling, and could serve as a refinement of CMDPO.

We provided some experimental results in Table 8. As discussed, although using `std` for normalization appears reasonable, it often underperforms compared to fixed scaling (`F_norm = 1`) due to lack of proper boundary constraints. Our proposed `clip_std` improves average performance by 2.4% over `std`, providing a more stable and effective normalization method.

## A.8 PROMPT AND REWARD FUNCTION

### A.8.1 PROMPT FOR REASONING.

Clear guiding instructions are added to the system prompt. These instructions encourage the model to output intermediate reasoning steps when generating answers, following a specific format to enhance its logical reasoning ability. Below is a related example.

### A.8.2 REWARD FUNCTION.

For most tasks, accuracy and formatting reward functions are employed. The Intersection over Union (IoU) reward function is applied for basic tasks.

> **SYSTEM:**
> A conversation between User and Assistant. The user asks a question, and the Assistant solves it. The assistant first thinks about the reasoning process in the mind and then provides the user with the answer. The reasoning process and answer are enclosed within **<think> </think>** and **<answer> </answer>** tags, respectively, i.e., **<think>** reasoning process here **</think><answer>** answer here **</answer>**

Figure 6: System Prompt for Reasoning.

Table 9: Comparison with PPO and DPO on mathematical reasoning.

| Methods | Avg | AIME24 | MATH-500 | AMC23 | Minerva | OlympiadBench |
|---|---|---|---|---|---|---|
| Still-3-1.5B-Preview | 51.6 | 32.5 | 84.4 | 66.7 | 29.0 | 45.4 |
| + DPO | 50.7 | 26.7 | 84.4 | 62.9 | 31.6 | 47.9 |
| + PPO | 52.9 | 33.3 | 82.8 | 70.0 | 29.4 | 48.9 |
| CMDPO-RS1 | 57.0 | 36.7 | 88.0 | 77.5 | 29.4 | 53.3 |
| CMDPO-RS3 | 57.1 | 36.7 | 85.4 | 80.0 | 29.0 | 54.2 |

Table 10: Comparison with PPO and DPO on multimodal classification.

| Methods | Avg | Flower102 | Pets37 | FGVC | Cars196 |
|---|---|---|---|---|---|
| Qwen2-VL-2B | 56.0 | 54.8 | 66.4 | 45.9 | 56.8 |
| + DPO | 73.7 | 68.4 | 75.9 | 66.8 | 83.6 |
| + PPO | 80.9 | 70.4 | 84.1 | 75.8 | 93.6 |
| + CMDPO | 88.3 | 77.1 | 88.0 | 90.8 | 97.3 |

- Accuracy: A reward of 1.0 is given if the model output matches the ground truth.

- Formatting: A reward of 1.0 is assigned if the required format "<think></think><answer></answer>" is followed.

- IoU: The reward is calculated based on the IoU score between the model-generated bounding box and the ground truth.

## A.9 COMPARISON WITH PPO AND DPO

We conduct supplementary experiments comparing CMDPO with both PPO and DPO. For fairness, all PPO and DPO models are trained with the same 7k-sample setting used in RS3.

**Mathematical Reasoning.** Table 9 reports results on six mathematical reasoning benchmarks. DPO slightly underperforms the base model (50.7 vs. 51.6), indicating instability under small-scale preference datasets. PPO shows marginal improvement (+1.3), but the gains are inconsistent. In contrast, CMDPO improves the average score by a large margin: +5.4 for RS1 and +5.5 for RS3, achieving the highest performance on AIME24, AMC23, and OlympiadBench. These results confirm that CMDPO provides a substantially more stable and effective optimization signal than PPO/DPO in the math reasoning domain.

**Multimodal Classification.** Table 10 presents results on five image classification datasets. DPO provides moderate gains (+17.7), while PPO offers stronger improvements (+24.9). However, CMDPO surpasses both by a large margin and reaches a new performance level (+32.3 over the base model). Notably, CMDPO achieves improvements of +30.9 (FGVC) and +40.5 (Cars196), showing that its KL-normalized update rule scales effectively in high-precision, fine-grained classification settings where PPO and DPO plateau.

Table 11: Comparison with PPO and DPO on reasoning grounding.

| Methods | mIoU_test | mIoU_val | gIoU_test |
|---|---|---|---|
| Qwen2-VL-2B | 26.9 | 30.1 | 25.3 |
| + DPO | 31.3 | 33.0 | 28.4 |
| + PPO | 36.8 | 33.8 | 35.2 |
| + CMDPO | 51.8 | 53.1 | 50.4 |

Table 12: Comparison of CMDPO and existing RLHF methods on the 7B-scale Qwen2.5-Math-7B model.

| Methods | Avg | AIME24 | MATH-500 | AMC23 | Minerva | OlympiadBench |
|---|---|---|---|---|---|---|
| Qwen2.5-Math-7B-Instruct | 43.8 | 13.3 | 79.8 | 50.6 | 34.6 | 40.7 |
| rStar-Math-7B | - | 26.7 | 78.4 | 47.5 | - | 47.1 |
| Eurus-2-7B-PRIME | 48.9 | 26.7 | 79.2 | 57.8 | 38.6 | 42.1 |
| Oat-Zero-7B | 47.8 | 30.0 | 80.6 | 55.4 | 29.0 | 44.0 |
| OpenReasoner-Zero-7B @ 8k | 45.9 | 13.3 | 82.4 | 54.2 | 31.6 | 47.9 |
| SimpleRL-Zero-7B | 46.6 | 26.7 | 78.2 | 60.2 | 27.6 | 40.3 |
| Qwen2.5-Math-7B | 30.9 | 13.3 | 57.6 | 45.0 | 14.7 | 23.7 |
| + DPO | 36.0 | 16.7 | 64.6 | 50.6 | 22.1 | 26.1 |
| + PPO | 46.9 | 26.7 | 78.4 | 62.5 | 29.8 | 37.3 |
| + GRPO | 43.7 | 16.7 | 73.4 | 62.5 | 30.2 | 35.7 |
| + GPG | 47.8 | 30.0 | 75.0 | 62.5 | 33.1 | 38.2 |
| + Dr. GRPO | 43.7 | 26.7 | 74.6 | 50.0 | 30.1 | 37.3 |
| + MDPO | 41.1 | 23.3 | 73.4 | 50.0 | 26.8 | 31.9 |
| + CMDPO | **51.2** | **36.7** | **82.8** | 62.9 | **34.2** | 39.3 |

**Reasoning Grounding.** Table 11 shows results on grounding tasks in terms of mIoU and gIoU. PPO yields significant gains (e.g., +9.9 gIoU), consistent with its known strength in continuous control and dense prediction tasks. DPO again shows modest improvements. CMDPO, however, dramatically outperforms both, with +24.9 mIoU (test) and +25.1 gIoU (test) over the base model, indicating that CMDPO's normalized KL-guided update provides a more stable and informative optimization signal in pixel-level tasks. This further supports the claim that CMDPO generalizes across reward scales and task modalities.

Across all three tasks, CMDPO consistently delivers the strongest performance and exhibits more stable optimization behavior than PPO and DPO. DPO shows signs of instability under limited preference data, and PPO provides moderate improvements but falls short of CMDPO, especially on fine-grained classification and segmentation tasks. These results directly address the reviewer's concern and demonstrate that CMDPO is not only competitive but substantially outperforms mainstream baselines when trained under identical conditions.

## A.10 EVALUATION ON 7B MODELS

To examine the scalability of CMDPO, we further evaluate it on the `Qwen2.5-Math-7B` model and compare it against a broader set of competitive baselines, including Dr. GRPO Liu et al. (2025c), Oat-Zero-7B Liu et al. (2025a), OpenReasoner-Zero-7B@8k Hu et al. (2025), and SimpleRL-Zero-7B Zeng et al. (2025).

Table 12 shows that CMDPO achieves the best overall performance among all RLHF baselines. CMDPO attains an average score of 51.2, outperforming PPO (+4.3), GPG (+3.4), and GRPO (+7.5). It also surpasses reasoning-specialized 7B models such as Eurus PRIME and Oat Zero, demonstrating that CMDPO remains effective and stable even at the 7B scale. Furthermore, CMDPO delivers substantial gains on the most challenging benchmarks. The largest improvements occur on AIME24 (+10.0 over PPO) and MATH-500 (+4.4 over GPG).

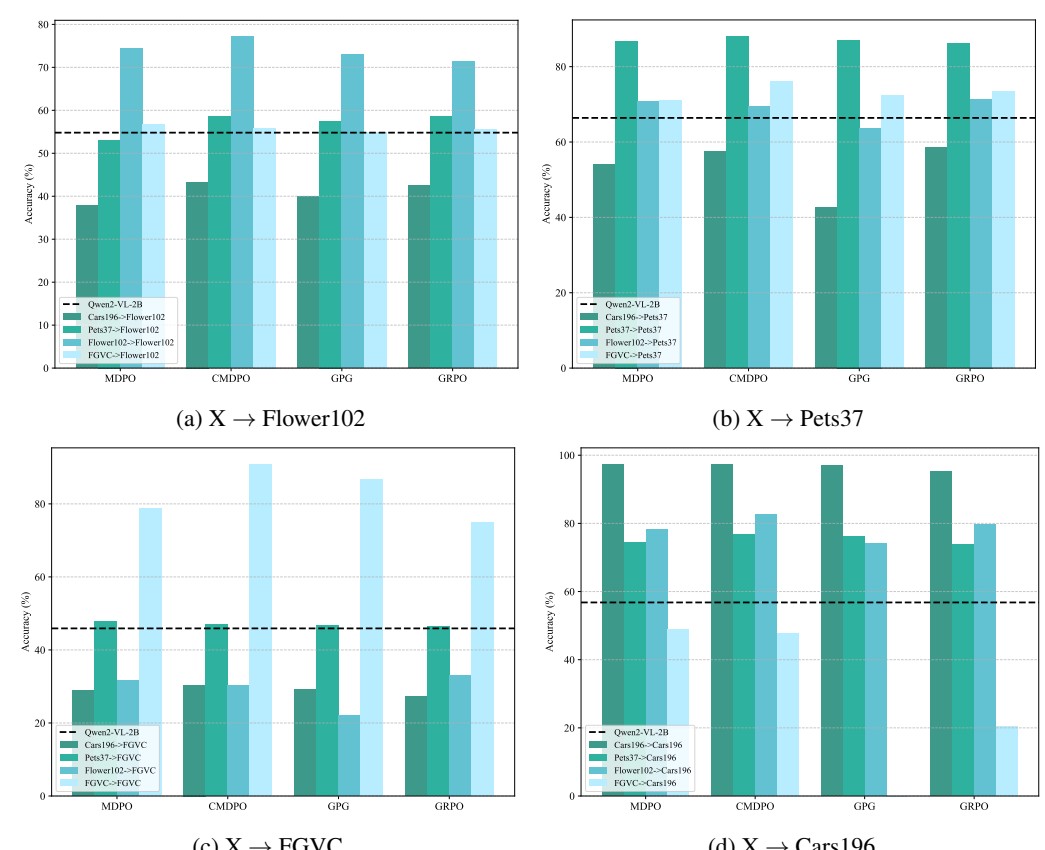

(a) X → Flower102

(b) X → Pets37

(c) X → FGVC

(d) X → Cars196

Figure 7: Generalization performance of different methods.

Table 13: Statistics of four classification datasets.

| Dataset | Flower102 | Pets37 | FGVC | Cars196 |
|---|---|---|---|---|
| Number | 408 | 148 | 400 | 784 |
| Categories | 102 | 37 | 100 | 196 |

## A.11 DETAILED GENERALIZATION ANALYSIS

As shown in Figure 7, CMDPO demonstrates superior generalization performance compared to other methods. GPG exhibits significant instability, achieving zero accuracy on the FGVC→Cars196 generalization task. We also observe that training on different datasets leads to varying generalization performance. Table 13 presents each dataset's data size and class distribution; under the 4-shot setting, each class contains four samples. The results indicate that higher data quantity does not necessarily lead to better generalization. Instead, reinforcement learning with a small amount of data can achieve comprehensive improvements while preserving the model's existing knowledge. This phenomenon may also be related to the inherent difficulty of the datasets, which we leave for future investigation.

## A.12 DETAILED LOSS ANALYSIS

As shown in the loss curves of Figure 8, the GPG method exhibits some degree of convergence, albeit with notable instability—for instance, a sharp drop in loss is observed during training on the Pets37 dataset. Conversely, CMDPO consistently achieves lower loss values than MDPO, with a particularly pronounced gap on the FGVC dataset, which is also reflected in the performance

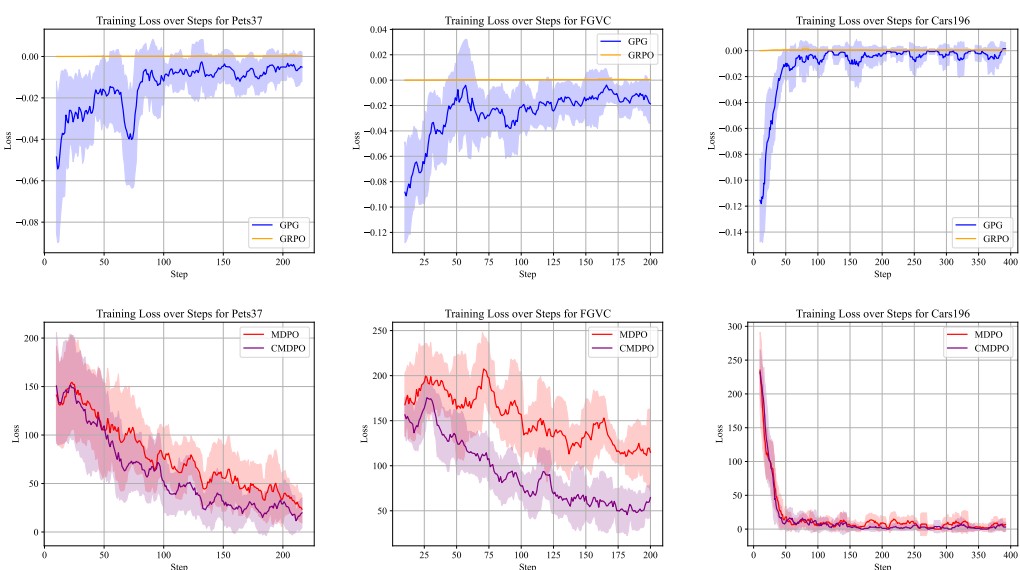

Figure 8: Loss curve on all classification datasets.

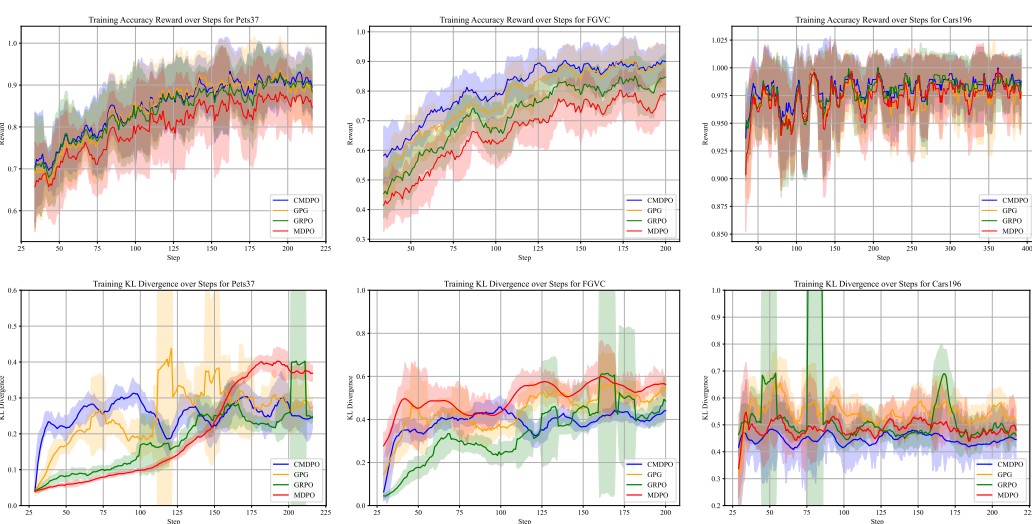

Figure 9: Comparison of smoothed training result reward and KL divergence on all classification datasets.

evaluation. On the Cars dataset, all three methods—GPG, MDPO, and CMDPO—converge well and perform comparably, slightly outperforming GRPO.

## A.13 DETAILED REWARD AND KL DIVERGENCE ANALYSIS

As shown in Figure 9, CMDPO maintains a high KL-reward conversion efficiency on the Pets37, FGVC, and Cars196 datasets, and the reward curves across different methods remain largely consistent. We also observe that GRPO and GPG exhibit instability in KL divergence, with sudden increases in both magnitude and variance during training. However, GRPO eventually converges to a KL divergence close to CMDPO's. Furthermore, CMDPO shows a distinct pattern in the KL curve: a rapid initial increase followed by a slower growth or even a slight decrease in later stages. This suggests that CMDPO encourages exploration of new reasoning paths in early training while preventing knowledge forgetting in later stages. In contrast, MDPO does not consistently exhibit

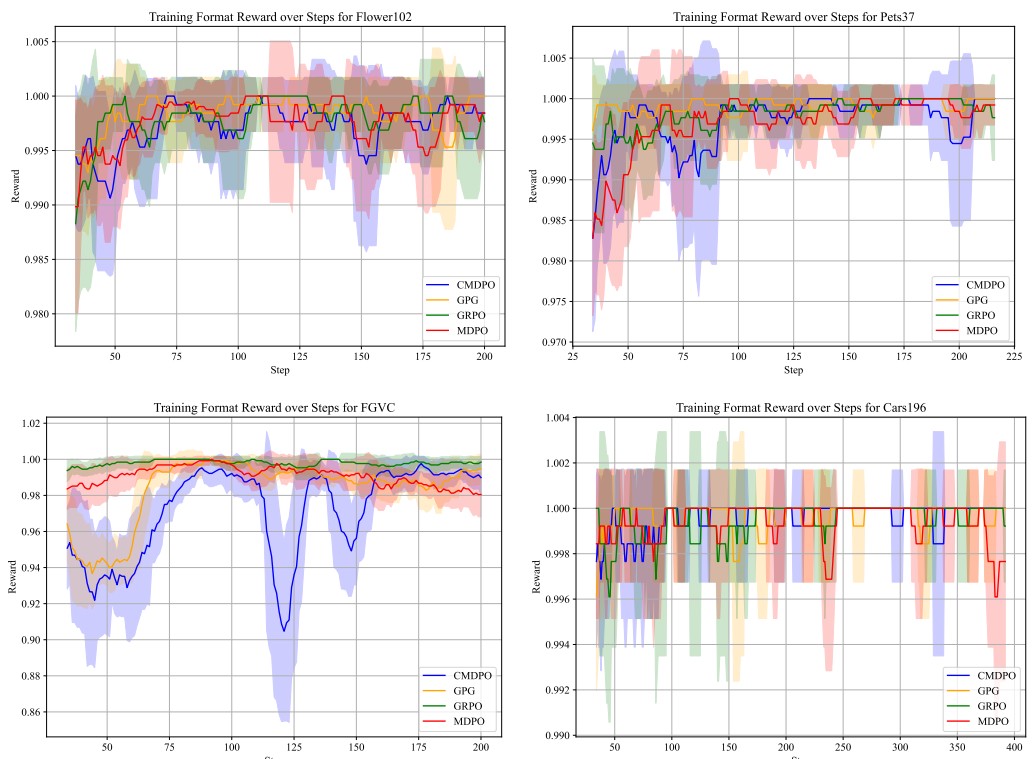

Figure 10: Smoothed training format reward curve.

Table 14: Performance of different RL variants with group normalization (GN). PG = policy gradient, IS = importance sampling, CLIP = clipping, F_norm = normalization factor in GN.

| Methods | Avg | Flower102 | Pets37 | FGVC | Cars196 |
|---|---|---|---|---|---|
| Qwen2-VL-2B | 56.0 | 54.8 | 66.4 | 45.9 | 56.8 |
| + GRPO (PG + IS + CLIP + $k_3$ + GN) | 81.9 | 71.4 | 86.1 | 74.8 | 95.3 |
| + GPG (PG + GN + $k_3$, F_norm = std) | 82.9 | 70.8 | 83.7 | 85.9 | 91.2 |
| + GPG (PG + GN, F_norm = std) | 84.3 | 72.1 | 85.3 | 84.8 | 95.1 |
| + GPG (PG + GN, F_norm = 1) | 86.0 | 73.0 | 87.1 | 86.8 | 97.1 |
| + GRPO (PG + IS + CLIP + $k_2$ + GN) | 80.7 | 70.2 | 85.3 | 73.7 | 93.6 |
| + GRPO (PG + IS + CLIP + $k_2^{\text{norm}}$ + GN) | 84.4 | 73.4 | 86.7 | 81.2 | 96.2 |
| + MDPO (PG + $k_2$ + GN) | 84.3 | 74.4 | 86.8 | 78.8 | 97.3 |
| + CMDPO (PG + $k_2^{\text{norm}}$ + GN) | 88.3 | 77.1 | 88.0 | 90.8 | 97.3 |

such behavior. The Pets37 dataset shows the opposite trend, leading to a sharp increase in KL divergence in the late stages and a significant deviation from the reference model.

Furthermore, we explored the changes in format rewards during training, as shown in Figure 10. Across all datasets, various methods quickly achieve high format reward scores, which increase much faster than result rewards. If both reward types are treated equally, the rapid saturation of format rewards can detract from the model's focus on improving result rewards. To address this, the proposed CMDPO employs a dynamic reward weighting strategy to balance the two types of rewards. During training, CMDPO exhibits a noticeable decrease in format rewards, allowing the model to explore more opportunities for enhancing result rewards.

Table 15: GRPO performance under different batch sizes

| batch_size | Avg | AIME24 | MATH-500 | AMC23 | Minerva | OlympiadBench |
|---|---|---|---|---|---|---|
| 1 | 49.9 | 26.7 | 83.8 | 64.9 | 28.6 | 45.4 |
| 2 | 49.6 | 23.3 | 85.8 | 66.7 | 29.0 | 43.3 |
| 4 | 50.8 | 33.3 | 84.4 | 62.9 | 27.9 | 45.4 |
| 8 | 50.4 | 26.7 | 83.2 | 62.9 | 29.0 | 50.2 |
| 16 | 50.3 | 33.3 | 83.2 | 62.9 | 29.0 | 43.3 |

Table 16: CMDPO performance under different batch sizes

| batch_size | Avg | AIME24 | MATH-500 | AMC23 | Minerva | OlympiadBench |
|---|---|---|---|---|---|---|
| 1 | 53.4 | 36.7 | 83.8 | 67.5 | 28.3 | 50.9 |
| 2 | 53.3 | 33.3 | 84.4 | 70.0 | 29.0 | 49.6 |
| 4 | 53.8 | 36.7 | 85.4 | 66.7 | 27.9 | 52.4 |
| 8 | 54.1 | 36.7 | 86.0 | 67.5 | 27.9 | 52.4 |
| 16 | 53.1 | 33.3 | 85.4 | 67.5 | 29.0 | 50.2 |

### A.14 COMPARISON OF RL VARIANTS WITH GROUP NORMALIZATION

We analyze several common RL algorithms within the standard policy gradient framework (see Appendix A.4 and A.5) and present the experimental results in Table 14. From the perspective of gradient estimation, the main differences among these algorithms lie in optimization techniques such as importance sampling (IS), CLIP, rule-based rewards, group normalization (GN), and KL divergence estimation. In particular, the use of importance sampling and clipping imposes constraints that influence training performance. Furthermore, CMDPO's $k_2^{\text{norm}}$ variant shows superior empirical performance compared to other KL divergence estimation methods in practice.

### A.15 IMPACT OF BATCH SIZE

To study the impact of batch size on GRPO and CMDPO in mathematical reasoning tasks, we randomly sampled 1000 test examples. The reward was computed using the cosine reward under the RS3 setting. Experimental results are shown in Table 15 and Table 16.

From the experimental results, it can be observed that both CMDPO and GRPO achieve optimal performance at moderate batch sizes, while extremely small or large batch sizes can slightly degrade performance. Based on prior studies Sutskever et al. (2014); Glorot & Bengio (2010); Srivastava et al. (2014), the possible reasons are as follows:

- Small batch sizes may lead to unstable convergence and oscillations during training;

- Large batch sizes can accelerate training but may cause the model to overlook some sample information, affecting performance.

The experiments indicate that CMDPO and GRPO are not highly sensitive to batch size. In addition, these experiments only varied batch size as a single variable; some optimizers may require higher learning rates with larger batch sizes to maintain the same convergence speed. Therefore, in practice, if sufficient computational resources are available, using a larger batch size with an appropriately increased learning rate is recommended, as it allows faster training without degrading performance.

## B ALGORITHM AND COMPLEXITY ANALYSIS OF SHARED SEGMENT DETECTION IN TDL

The core of shared segment detection in TDL is a longest common substring matching problem. We describe an algorithm for shared segment detection (Algorithm 1) implemented using *difflib*.

---

**Algorithm 1** Shared Segment Detection with *difflib*

---

**Require:** Sequences $S = \{s_1, s_2, \ldots, s_k\}$, minimum length $x$
**Ensure:** Set $\mathcal{C}_{\text{shared}}$ of segments appearing in all $s_i$
1: $\mathcal{C}_{\text{shared}} \leftarrow$ all substrings of $s_1$ (implicitly represented)
2: **for** $i = 2$ **to** $k$ **do**
3:     $\mathcal{C}_i \leftarrow \emptyset$
4:     /* Compute matching blocks between $s_1$ and $s_i$ using a SequenceMatcher-like procedure */
5:     Use a matching-block finder to obtain blocks $(p, q, \ell)$ where $s_1[p : p + \ell] = s_i[q : q + \ell]$ and
       $\ell \geq x$
6:     For each such block $(p, q, \ell)$, add substring $s_1[p : p + \ell]$ to $\mathcal{C}_i$
7:     $\mathcal{C}_{\text{shared}} \leftarrow \mathcal{C}_{\text{shared}} \cap \mathcal{C}_i$
8: **end for**
9: **return** $\mathcal{C}_{\text{shared}}$

---

**Algorithm 2** Shared Segment Detection with Rolling Hash + Binary Search

---

**Require:** A set of $k$ token sequences $S = \{s_1, s_2, \ldots, s_k\}$, minimum segment length $x \in \mathbb{N}^+$
**Ensure:** Set $\mathcal{C}_{\text{shared}}$ of segments appearing in all $s_i$ with length $\geq x$
1: Define **hash_set**(seq, L) as the set of rolling hashes of all length-$L$ substrings in seq
2: $L_{\min} \leftarrow x$, $L_{\max} \leftarrow \min_i |s_i|$
3: $\mathcal{C}_{\text{shared}} \leftarrow \emptyset$
4: **while** $L_{\min} \leq L_{\max}$ **do**
5:     $L \leftarrow \lfloor (L_{\min} + L_{\max})/2 \rfloor$
6:     $H \leftarrow$ hash_set$(s_1, L)$
7:     **for** $i = 2$ **to** $k$ **do**
8:         $H \leftarrow H \cap$ hash_set$(s_i, L)$
9:         **if** $H = \emptyset$ **then**
10:           **break**
11:         **end if**
12:     **end for**
13:     **if** $H \neq \emptyset$ **then**
14:         Update $\mathcal{C}_{\text{shared}}$ with substrings corresponding to hashes in $H$
15:         $L_{\min} \leftarrow L + 1$
16:     **else**
17:         $L_{\max} \leftarrow L - 1$
18:     **end if**
19: **end while**
20: **return** $\mathcal{C}_{\text{shared}}$

---

**Time Complexity.** The nested loops compare every starting position in $s_1$ with every starting position in $s_i$, leading to average time $O(|s_1| \cdot |s_i|)$ per pair $(s_1, s_i)$. Summing over $i = 2$ to $k$, the total time is

$$\sum_{i=2}^{k} O(n_1 n_i) = O\left(n_1 \sum_{i=2}^{k} n_i\right),$$

where $n_j = |s_j|$. Let $n$ be the average sequence length (i.e., $n_j = O(n)$). Then the overall complexity is $O(kn^2)$. Due to limited computational resources, both $k$ and $n$ are kept small in our experimental setup, resulting in very low runtime overhead for this algorithm, typically around one second.

**Algorithm Optimization.** It is worth noting that the above algorithm is a naive implementation. Depending on the specific scenario, more advanced string-matching algorithms can be employed to handle larger values of $k$ and $n$. For instance, one could use Rolling Hash combined with binary search (average time complexity $O(kn \log n)$), or more sophisticated data structures such as a Generalized Suffix Automaton (GSAM) or Generalized Suffix Tree (average time complexity $O(kn)$).

Among these alternatives, we present the Rolling Hash + binary search approach, as it is the easiest to implement (Algorithm 2).

Table 17: Performance impact of DRW and TDL across different RL methods.

| Methods | Avg | Flower102 | Pets37 | FGVC | Cars196 |
|---|---|---|---|---|---|
| Qwen2-VL-2B | 56.0 | 54.8 | 66.4 | 45.9 | 56.8 |
| CMDPO | 87.2 | 76.3 | 87.5 | 88.5 | 96.3 |
| + DRW | 87.9 | 76.7 | 88.2 | 89.9 | 96.9 |
| + TDL | 87.6 | 76.0 | 87.2 | 90.0 | 97.1 |
| GRPO | 81.9 | 71.4 | 86.1 | 74.8 | 95.3 |
| + DRW | 82.0 | 71.2 | 86.6 | 75.2 | 95.1 |
| + TDL | 82.4 | 71.6 | 86.8 | 75.8 | 95.3 |
| MDPO | 84.3 | 74.4 | 86.8 | 78.8 | 97.3 |
| + DRW | 85.1 | 74.6 | 87.3 | 81.4 | 97.3 |
| + TDL | 84.8 | 75.2 | 87.0 | 80.0 | 97.0 |

Table 18: Effect of $\tau$ on classification accuracy.

| $\tau$ | Avg | Flower102 | Pets37 | FGVC | Cars196 |
|---|---|---|---|---|---|
| 0.7 | 85.5 | 75.8 | 83.7 | 88.0 | 94.5 |
| 0.8 | 87.5 | 76.2 | 87.0 | 89.7 | 96.9 |
| 0.9 | 88.3 | 77.1 | 88.0 | 90.8 | 97.3 |
| 1.0 | 88.7 | 77.8 | 88.4 | 91.0 | 97.5 |
| 1.1 | 88.5 | 77.5 | 88.0 | 90.8 | 97.6 |

The algorithm applies a binary search on the possible segment length $L$ over the interval $[x, \min_i |s_i|]$, which requires $O(\log n)$ iterations. For each candidate length $L$, we compute the rolling hash of all length-$L$ substrings for every sequence. The number of such substrings in a sequence of length $|s_i|$ is $O(n)$, and each substring hash can be computed in $O(1)$ time using a rolling hash, resulting in $O(n)$ time per sequence. For $k$ sequences, the total cost for hash computation is $O(kn)$.

After computing the hashes, we take the intersection of the $k$ hash sets to determine the common substrings of length $L$. Assuming the hash set size is $O(n)$, the intersection over $k$ sequences can be computed in $O(kn)$ time.

Combining the binary search and per-length computation, the overall time complexity of the algorithm is $O(kn \log n)$.

Compared with the naive approach of comparing all pairs of substrings which has $O(kn^2)$ complexity, this method is significantly more efficient. Further optimizations include using hash tables to reduce constant factors in intersection operations, and pruning search using the shortest sequence when $L$ is large.

## C  ANALYSIS OF OPTIONAL STABILIZATION HEURISTICS

### C.1  COMMONALITY ANALYSIS

We evaluate whether DRW and TDL serve as general stabilization modules across RL methods, and whether CMDPO remains strong without them. As shown in Table 17, CMDPO already achieves the highest base performance, indicating that its centered update yields stable optimization without relying on auxiliary heuristics. Incorporating DRW or TDL provides small but consistent gains. For GRPO and MDPO, both heuristics lead to more noticeable improvements, particularly on FGVC and Cars196.

Table 19: Effect of $s$ on classification accuracy.

| $s$ | Avg | Flower102 | Pets37 | FGVC | Cars196 |
|---|---|---|---|---|---|
| 6 | 85.9 | 74.5 | 86.6 | 86.4 | 95.9 |
| 8 | 87.7 | 76.6 | 87.3 | 90.1 | 97.0 |
| 10 | 88.3 | 77.1 | 88.0 | 90.8 | 97.3 |
| 12 | 88.7 | 77.6 | 88.9 | 90.9 | 97.4 |
| 14 | 87.9 | 77.4 | 87.2 | 90.0 | 96.8 |

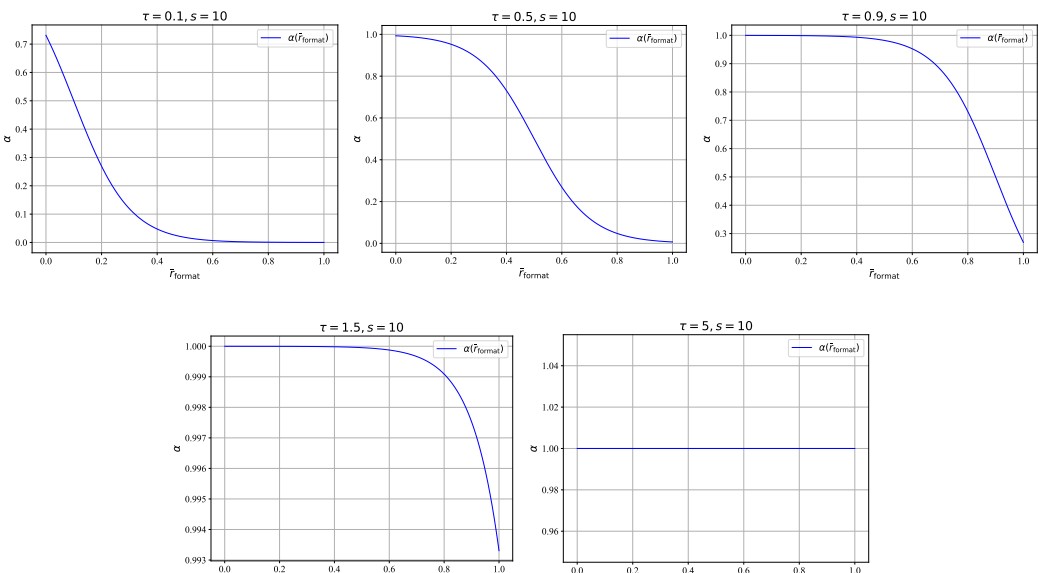

Figure 11: Dynamic reward function visualization with varying $\tau$.

## C.2 HYPERPARAMETER ANALYSIS IN DRW

We conduct extensive experiments to examine the hyperparameters of DRW and provide practical guidance for their selection. As discussed in Section 4.4 (Table 4), the decay factor $\delta$ controls the attenuation strength in TDL. Since $\delta \in [0, 1)$ determines how aggressively common segments are down-weighted, any non-trivial value generally leads to performance gains.

DRW is designed to penalize premature overfitting to formatting rewards, ensuring that the policy does not focus excessively on structural patterns while ignoring the target answers. Its two hyperparameters, $(s, \tau)$, play distinct roles: $\tau$ determines the position of the decay onset, where a smaller $\tau$ introduces decay at lower reward values, and $s$ controls the sharpness of the decay. We present the performance trends under varying $s$ and $\tau$ in Tables 18 and 19.

For $\tau$, although it is not bounded in theory, very small values cause a sharp performance drop because the decay is activated too early, overly penalizing the formatting reward before the model has learned the desired structure. Extremely large $\tau$ values suppress decay entirely, effectively removing DRW (the decay curves are provided in Figures 11 and 12). We recommend setting $\tau$ within the range $1 \pm 0.1$.

For $s$, very small values lead to an almost linear decay and impose excessive penalties too early, whereas very large values delay the decay until the reward approaches $\tau$. When $s$ becomes sufficiently large, the decay curve converges toward the identity function $y = 1$, plummeting to 0 at $\tau$.

Overall, the experimental results suggest that the default setting of $(s, \tau)$ provides a stable and reliable choice across tasks.

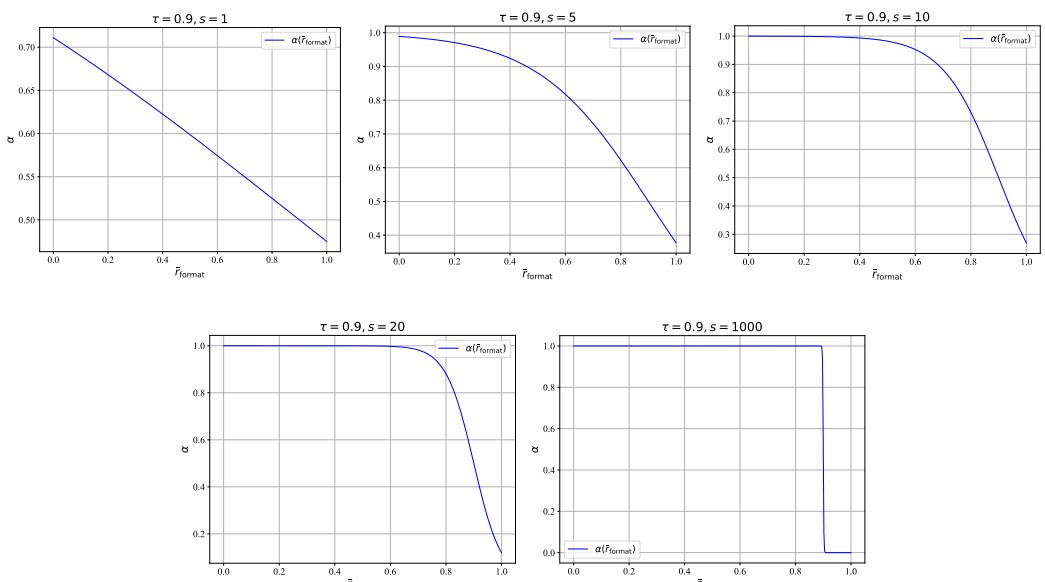

Figure 12: Dynamic reward function visualization with varying $s$.

## D  THE USE OF LARGE LANGUAGE MODELS (LLMS)

We thank Qwen3 Plus for its assistance in the writing and language polishing of this paper.

