# OpenReview forum: "CMDPO: Centered Mirror Descent Policy Optimization for Stable and Efficient Reinforcement Learning"
_ICLR.cc/2026/Conference — Submitted to ICLR 2026_

### Official Review · Reviewer_gTMg · 2025-10-30

**Soundness:** 2
**Presentation:** 2
**Contribution:** 2
**Rating:** 4
**Confidence:** 4

**Summary:**

This paper introduces Centered Mirror Descent Policy Optimization (CMDPO), a novel reinforcement learning algorithm for aligning large language models. It aims to overcome limitations of existing methods like PPO, DPO, and MDPO. CMDPO's core innovation is a group centering technique that mathematically eliminates the need to estimate the problematic partition function (log Z) present in MDPO, thereby achieving unbiased and theoretically sound policy updates. The framework is further enhanced with dynamic reward weighting to balance task and format rewards and token-level discriminative learning to focus on quality-indicative tokens.

**Strengths:**

1. CMDPO directly addresses a theoretical weakness in MDPO (partition function estimation bias) with a mathematically solution that provides unbiased estimates and guarantees a unique optimal solution.
2. The dynamic reward weighting and token-level discriminative learning mechanisms address practical challenges in RL training, such as balancing different reward signals and improving fine-grained learning focus.

**Weaknesses:**

1. The overall contribution might be seen as an incremental improvement over MDPO, combined with practical heuristics (dynamic weighting, token-level learning) that are potentially applicable to other RL algorithms.
2. The framework introduces several new components and associated hyperparameters (e.g., $s$, $\tau$ for dynamic weighting; $\delta$ for token learning), potentially making tuning more complex than simpler methods like GRPO or DPO.

**Questions:**

See Weaknesses

---

> ### Author Response · Authors · 2025-11-20
> **Part 1/2**
>
> > **W1.** The overall contribution might be seen as an incremental improvement over MDPO, combined with practical heuristics (dynamic weighting, token-level learning) that are potentially applicable to other RL algorithms.
>
> **Ans:**
>
> As noted by reviewers Z5Tq, snYZ, and zrs1 in the **Strengths** section, our work **addresses the logZ estimation problem in an elegant and principled manner**, with **strong theoretical guarantees**. The proposed CMDPO is **simple yet effective**, achieving superior empirical performance.
>
> To better highlight the core contribution of this work, we have made the following revisions **according to Z5Tq's comment W2**. We now present **CMDPO as the central method**—emphasizing its theoretical foundation, including guarantees of convergence and unbiased estimation—and have strengthened its conceptual presentation in both the introduction and method sections. CMDPO stands as a self-contained and effective algorithm that outperforms existing methods even without the use of DRW or TDL.
>
> Meanwhile, we have repositioned **DRW and TDL as optional heuristic enhancements**. These components aim to further improve reward stability and enable finer-grained optimization toward high-quality outputs, and are designed to be generally applicable to other policy optimization frameworks.
>
> In the experiments, we report results for CMDPO alone, as well as incremental improvements when DRW and TDL are added, clearly separating the contributions of the core method from those of the enhancements. We also apply DRW and TDL to other RL methods such as GRPO and MDPO to demonstrate their generalizability.
>
> We believe these revisions more effectively emphasize the theoretical contribution of CMDPO, while fairly showcasing the practical value of DRW and TDL without compromising the clarity or independence of the core approach.
>
> | Models | Average | Flower102 | Pets37 | FGVC | Cars196 |
> |---|:---:|:---:|:---:|:---:|:---:|
> | Qwen2-VL-2B | 56.0  | 54.8  | 66.4  | 45.9  | 56.8  |
> |  CMDPO | 87.2  | 76.3  | 87.5  | 88.5  | 96.3  |
> | + DRW  | 87.9  | 76.7  | 88.2  | 89.9  | 96.9  |
> | + TDL | 87.6  | 76.0  | 87.2  | 90.0  | 97.1  |
> | GRPO | 81.9  | 71.4  | 86.1  | 74.8  | 95.3  |
> | + DRW  | 82.0  | 71.2  | 86.6  | 75.2  | 95.1  |
> | + TDL | 82.4  | 71.6  | 86.8  | 75.8  | 95.3  |
> | MDPO | 84.3  | 74.4  | 86.8  | 78.8  | 97.3  |
> | + DRW  | 85.1  | 74.6  | 87.3  | 81.4  | 97.3  |
> | + TDL | 84.8  | 75.2  | 87.0  | 80.0  | 97.0  |

---

> ### Author Response · Authors · 2025-11-20
> **Part 2/2**
>
> > **W2.** The framework introduces several new components and associated hyperparameters (e.g., $s$, $\tau$ for dynamic weighting; $\delta$ for token learning), potentially making tuning more complex than simpler methods like GRPO or DPO.
>
> **Ans:**
>
> DRW balances format and task rewards, while TDL focuses on difference learning. Both only affect the computation of the actual and implicit rewards, and **do not impact the theoretical guarantees of CMDPO**. Moreover, DRW and TDL are optional heuristic components; CMDPO still outperforms other methods even without them.
>
> | Models | Average | Flower102 | Pets37 | FGVC | Cars196 |
> |---|:---:|:---:|:---:|:---:|:---:|
> | Qwen2-VL-2B | 56.0  | 54.8  | 66.4  | 45.9  | 56.8  |
> |  CMDPO | 87.2  | 76.3  | 87.5  | 88.5  | 96.3  |
> | + DRW  | 87.9  | 76.7  | 88.2  | 89.9  | 96.9  |
> | + TDL | 87.6  | 76.0  | 87.2  | 90.0  | 97.1  |
> | GRPO | 81.9  | 71.4  | 86.1  | 74.8  | 95.3  |
> | + DRW  | 82.0  | 71.2  | 86.6  | 75.2  | 95.1  |
> | + TDL | 82.4  | 71.6  | 86.8  | 75.8  | 95.3  |
> | MDPO | 84.3  | 74.4  | 86.8  | 78.8  | 97.3  |
> | + DRW  | 85.1  | 74.6  | 87.3  | 81.4  | 97.3  |
> | + TDL | 84.8  | 75.2  | 87.0  | 80.0  | 97.0  |
>
> Regarding TDL, **we have already analyzed the impact of its hyperparameter $\delta$ in Table 4 of the submitted manuscript**. As $\delta$ controls the degree to which contributions from common prefixes are down-weighted, it is defined in the range [0,1). Our results show that TDL consistently brings performance gains as long as $\delta$ is not set too close to zero.
>
> DRW is designed to penalize premature overfitting to format patterns by applying a dynamic penalty on format rewards, thereby encouraging the policy to explore toward semantically correct answers. It introduces two hyperparameters: $\tau$, which determines the onset of reward decay, and $s$, which controls the steepness of the decay. Below is an ablation study analyzing their effects:
>
> | $\tau$ | Average | Flower102 | Pets37 | FGVC | Cars196 |
> |:---:|:---:|:---:|:---:|:---:|:---:|
> | 0.7 | 85.5  | 75.8  | 83.7  | 88.0  | 94.5  |
> | 0.8 | 87.5  | 76.2  | 87.0  | 89.7  | 96.9  |
> | 0.9 | 88.3  | 77.1  | 88.0  | 90.8  | 97.3  |
> | 1 | 88.7  | 77.8  | 88.4  | 91.0  | 97.5  |
> | 1.1 | 88.5  | 77.5  | 88.0  | 90.8  | 97.6  |
>
> | $s$ | Average | Flower102 | Pets37 | FGVC | Cars196 |
> |:---:|:---:|:---:|:---:|:---:|:---:|
> | 6 | 85.9  | 74.5  | 86.6  | 86.4  | 95.9  |
> | 8 | 87.7  | 76.6  | 87.3  | 90.1  | 97.0  |
> | 10 | 88.3  | 77.1  | 88.0  | 90.8  | 97.3  |
> | 12 | 88.7  | 77.6  | 88.9  | 90.9  | 97.4  |
> | 14 | 87.9  | 77.4  | 87.2  | 90.0  | 96.8  |
>
> For $\tau$, although theoretically unbounded, values that are too small cause the format reward to decay prematurely, hindering the model’s ability to learn proper formatting. Conversely, excessively large values make the decay ineffective, rendering DRW equivalent to no penalty at all (we include illustrative curves of the dynamic reward weighting function under different $\tau$ in the appendix for clarity). Empirically, we recommend setting $\tau$ within the range 1±0.1.
>
> For $s$, small values result in a nearly linear penalty that takes effect too early, while very large values delay the decay until the reward approaches $\tau$. When $s$ is sufficiently large, the weighting function approaches a constant (i.e., y=1), again nullifying the effect of DRW. Our experiments suggest that values between 10 and 12 offer a robust trade-off.
>
> As demonstrated in our response to W2, the default ($s$,$\tau$) settings achieve strong and consistent performance across multiple benchmarks with minimal tuning, making them practical for general use.

---

### Official Review · Reviewer_snYZ · 2025-10-31

**Soundness:** 3
**Presentation:** 2
**Contribution:** 3
**Rating:** 4
**Confidence:** 4

**Summary:**

The paper develops an interesting approach for post-training LLMs, improving upon MDPO by introducing a group-centering trick that removes the need to estimate the difficult partition function. It also adds dynamic reward weighting to automatically balance task and format rewards, and token-level discriminative learning to focus on meaningful tokens rather than repetitive text. Experimental results show promising improvements over standard baselines.

**Strengths:**

- One of the key and primary strength of the paper lies in the group centering idea which cleanly removes estimation of the log-partition and heavy computation thereby keeping MDPO’s convergence guarantees while improving training stability.
- The variance-covariance analysis part is ineteresting and tells why it stabilizes learning
- Dynamic Re-weighting and Token-level Discriminative Learning are simple but effective ideas
- The paper shows promising results by outperforming baselines in several math olympiad and other benchmarks.

**Weaknesses:**

- One of the most confusing part is that 2-3 components have been added like Dynamic Re-weighting and Token-level Discriminative Learning which I appreciate that improves the performance but is not clearly analyzed
-  Can a detailed plot be shown to understand the benefit of each and how and why each is helping over others?
-  Also these additional two methods come as adhoc to the story which makes it slightly confusing from the central theme that is improving the tractability of MDPO
- How does the batch-size affect the convergence is crucial? A clear study will be helpful to understand the batch-size and how it impacts in the CMDPO performance? How is the sensitivity compared to GRPO?
- In GRPO standard also, the same reward normalization is done but the learning objective is difference, so a clean analysis comparing the two with one factor at a time will be helpful.

I am interested to understand it better and will increase score accordingly.

**Questions:**

See weakness

---

> ### Author Response · Authors · 2025-11-20
> **Part 1/4**
>
> > **W1.** One of the most confusing part is that 2-3 components have been added like Dynamic Re-weighting and Token-level Discriminative Learning which I appreciate that improves the performance but is not clearly analyzed
>
> **Ans:**
>
> Refer to the reply to W3.
>
> > **W2.** Can a detailed plot be shown to understand the benefit of each and how and why each is helping over others?
>
> **Ans:**
>
> We sincerely thank the reviewer for this insightful suggestion. **The core contribution of our work is the CMDPO loss, which elegantly and principledly addresses the logZ estimation problem with strong theoretical guarantees.** DRW and TDL are optional, general heuristic enhancements that serve the CMDPO loss. For CMDPO, DRW provides a more **task-enhanced actual reward** by balancing format and task rewards in the reward space. Specifically, DRW dynamically suppresses premature saturation of format rewards, encouraging the policy model to explore more task-related reasoning paths. TDL, on the other hand, provides a **more efficient implicit reward** at a finer granularity. It reduces the influence of indistinguishable parts (common prefixes) in the generation path and focuses more on the key features that truly distinguish high-quality outputs. **Both components only affect the computation of the actual and implicit rewards and do not impact the theoretical guarantees of CMDPO.**
>
> To better illustrate these effects, we provide ablation studies and hyperparameter analysis on DRW and TDL. Please **refer to the response to W3** for details.

---

> > ### Author Response · Authors · 2025-11-20
> > **Part 3/4**
> >
> > > **W4.** How does the batch-size affect the convergence is crucial? A clear study will be helpful to understand the batch-size and how it impacts in the CMDPO performance? How is the sensitivity compared to GRPO?
> >
> > Due to limited computational resources, we randomly sampled 1,000 instances to analyze the impact of batch size, using the cosine reward setting from RS3.
> >
> > | GRPO | batch_size | Average | AIME24 | MATH-500 | AMC23 | Minerva | OlympiadBench |
> > |:---:|---|:---:|:---:|:---:|:---:|:---:|:---:|
> > |  | 1 | 49.9  | 26.7  | 83.8  | 64.9  | 28.6  | 45.4  |
> > |  | 2 | 49.6  | 23.3  | 85.8  | 66.7  | 29.0  | 43.3  |
> > |  | 4 | 50.8  | 33.3  | 84.4  | 62.9  | 27.9  | 45.4  |
> > |  | 8 | 50.4  | 26.7  | 83.2  | 62.9  | 29.0  | 50.2  |
> > |  | 16 | 50.3  | 33.3  | 83.2  | 62.9  | 29.0  | 43.3  |
> >
> > | CMDPO | batch_size | Average | AIME24 | MATH-500 | AMC23 | Minerva | OlympiadBench |
> > |:---:|---|:---:|:---:|:---:|:---:|:---:|:---:|
> > |  | 1 | 53.4  | 36.7  | 83.8  | 67.5  | 28.3  | 50.9  |
> > |  | 2 | 53.3  | 33.3  | 84.4  | 70.0  | 29.0  | 49.6  |
> > |  | 4 | 53.8  | 36.7  | 85.4  | 66.7  | 27.9  | 52.4  |
> > |  | 8 | 54.1  | 36.7  | 86.0  | 67.5  | 27.9  | 52.4  |
> > |  | 16 | 53.1  | 33.3  | 85.4  | 67.5  | 29.0  | 50.2  |
> >
> > The results show that both CMDPO and GRPO achieve optimal performance with moderate batch sizes, while performance degrades when the batch size is either too small or too large. Drawing from prior studies [1][2][3], a possible explanation is that overly small batch sizes may lead to unstable training or even oscillations during convergence, whereas excessively large batch sizes, although accelerating training, may cause the model to overlook fine-grained sample information, potentially harming generalization. However, the experiments indicate that CMDPO and GRPO are relatively insensitive to batch size variations. It is worth noting that in the above analysis, batch size was treated as the only variable; in practice, larger batch sizes often require higher learning rates to maintain the same convergence dynamics. **Therefore, when computational resources permit, we recommend using larger batch sizes with appropriately increased learning rates—this not only speeds up training but does not necessarily compromise performance.**
> >
> > [1] Sutskever, I., Vinyals, O., & Le, Q. (2014). Sequence to sequence learning with neural networks. arXiv preprint arXiv:1409.3215.
> >
> > [2] Glorot, X., & Bengio, Y. (2010). Understanding the difficulty of training deep feedforward neural networks. In Proceedings of the thirteenth international conference on artificial intelligence and statistics (pp. 249-256).
> >
> > [3] Srivastava, N., Hinton, G., Krizhevsky, A., Sutskever, I., & Erhan, D. (2014). Dropout: a simple way to prevent neural networks from overfitting. The Journal of Machine Learning Research, 15(2), 1929-1958.

---

> > > ### Comment · Reviewer_snYZ · 2025-11-28
> > > **Response to Rebuttal by Authors**
> > >
> > > Thank the authors for the detailed rebuttal and for addressing majority of my concerns. All my major concerns are addressed
> > > i appreciate the additional experimental ablations. I have also read the response to the other reviewers.
> > > However, I still believe Dynamic Re-weighting and Token-level Discriminative Learning are additional aspects and not the central theme which confuses the main theme of the paper. Can you pls briefly describe the connections clearly? Its still not very clear.

---

> > > > ### Author Response · Authors · 2025-11-28
> > > > **The connection between DRW, TDL, and the core CMDPO**
> > > >
> > > > Thank you for your feedback and valuable suggestions. **Following the second suggestion from Reviewer Z5Tq's Weakness section**, we have emphasized CMDPO as our core methodology, with **DRW and TDL serving as optional heuristic enhancements**. This emphasis ensures a clear focus in our paper (please refer to the revised PDF). Given that both the response and the paper thoroughly explain the roles and motivations of DRW and TDL, here we primarily **elucidate how these components assist CMDPO**.
> > > >
> > > > The relationship between CMDPO, DRW, and TDL can be **summarized** as follows:
> > > >
> > > > ```plaintext
> > > > TDL --(Enhances)--> CMDPO <--(Enhances)-- DRW
> > > > ```
> > > >
> > > > In our work, CMDPO's learning process is described as the progression from **an implicit reward space towards the true reward space**:
> > > >
> > > > ```plaintext
> > > > CMDPO: Implicit Reward --> True Reward
> > > > ```
> > > >
> > > > It's important to note that **DRW** enhances this process by balancing form rewards and task-specific rewards, thereby providing more robust **true rewards**. **TDL** aids by filtering out noise from irrelevant tokens in the **implicit reward** space, thus refining it. These relationships can be represented as:
> > > >
> > > > ```plaintext
> > > > TDL --(Enhances)--> Implicit Reward
> > > > DRW --(Enhances)--> True Reward
> > > > ```
> > > >
> > > > **Thus, the overall structure is illustrated below:**
> > > >
> > > > ```
> > > >                             +-------------------+
> > > >                             |       CMDPO       |
> > > >                             +-------------------+
> > > > TDL  ───(Noise Filter)────> |  Implicit Reward  |
> > > >                             |         |         |
> > > >                             |         v         |
> > > > DRW  ──(Rewards Balance)──> |    True Reward    |
> > > >                             +-------------------+
> > > > ```
> > > >
> > > >
> > > > We warmly welcome any further questions or suggestions from the reviewers. And during the ICLR discussion phase, we will respond promptly and do our best to improve the work.

---

> > ### Author Response · Authors · 2025-11-20
> > **Part 4/4**
> >
> > > **W5.** In GRPO standard also, the same reward normalization is done but the learning objective is difference, so a clean analysis comparing the two with one factor at a time will be helpful.
> >
> > This is a very interesting question. We provide some initial discussion and analysis in the appendix. We place several common RL algorithms—excluding DPO, which is strictly a preference learning method based on the Bradley-Terry model—within the classical policy gradient framework. Both MDPO and our CMDPO can be reformulated under this framework (see Appendix A.4 and A.5). **The key differences among these methods lie in various optimization techniques, including importance sampling, CLIP clipping, rule-based rewards, group normalization, and KL divergence estimation**. Notably, methods such as GRPO, GPG, MDPO, and CMDPO all employ rule-based rewards. Group normalization was introduced and analyzed in [1]. Regarding importance sampling and CLIP, [2] argues that reverting to the original policy gradient formulation can yield better performance, supported by empirical evidence. Therefore, our main focus is on the differences in KL divergence estimation (Appendix A.5 and A.6) and the role of the standard deviation (std) in group normalization (Appendix A.7). Unfortunately, due to computational costs and time constraints, we could only include theoretical analysis and limited empirical evaluation of KL divergence behavior in the initial submission.
> >
> > Fortunately, the ICLR discussion phase offers great flexibility, enabling us to present more detailed experimental analysis in this revision.
> >
> > | Models | Average | Flower102 | Pets37 | FGVC | Cars196 |
> > |---|:---:|:---:|:---:|:---:|:---:|
> > | Qwen2-VL-2B | 56.0  | 54.8  | 66.4  | 45.9  | 56.8  |
> > | + GRPO  (PG+IS+CLIP+K3 +GN) | 81.9  | 71.4  | 86.1  | 74.8  | 95.3  |
> > | + GPG (PG + GN+K3, F_norm   =std) | 82.9  | 70.8  | 83.7  | 85.9  | 91.2  |
> > | + GPG (PG + GN, F_norm =std) | 84.3  | 72.1  | 85.3  | 84.8  | 95.1  |
> > | + GPG (PG + GN, F_norm =1) | 86.0  | 73.0  | 87.1  | 86.8  | 97.1  |
> > | + GRPO (PG+IS+CLIP+K2 +GN) | 80.7  | 70.2  | 85.3  | 73.7  | 93.6  |
> > | + GRPO (PG+IS+CLIP+K2_norm  +GN) | 84.4  | 73.4  | 86.7  | 81.2  | 96.2  |
> > | + MDPO (PG+K2 +GN) | 84.3  | 74.4  | 86.8  | 78.8  | 97.3  |
> > | + CMDPO (PG+K2_norm +GN) | 88.3  | 77.1  | 88.0  | 90.8  | 97.3  |
> >
> > Here, PG denotes policy gradient, IS denotes importance sampling—typically combined with the clipping technique CLIP to ensure training stability—GN stands for group normalization, and F_norm refers to the denominator term in group normalization.
> >
> > | Models | Average | Flower102 | Pets37 | FGVC | Cars196 |
> > |---|:---:|:---:|:---:|:---:|:---:|
> > | Qwen2-VL-2B | 56.0  | 54.8  | 66.4  | 45.9  | 56.8  |
> > | + CMDPO (F_norm =1) | 88.3  | 77.1  | 88.0  | 90.8  | 97.3  |
> > | + CMDPO (F_norm =std) | 86.5  | 75.4  | 87.3  | 86.8  | 96.5  |
> > | + CMDPO (F_norm =clip_std) | 88.9  | 78.0  | 89.2  | 91.4  | 97.1  |
> >
> > Moreover, as discussed in Section A.7, **although normalization based on standard deviation (std) provides a seemingly reasonable scaling mechanism, it often underperforms compared to the case where F_norm = 1, due to the lack of proper boundary constraints.** In contrast, our proposed improvement, clip_std, achieves an average performance gain of 2.4% over the standard std approach.
> >
> > [1] Shao, Zhihong, et al. "Deepseekmath: Pushing the limits of mathematical reasoning in open language models." arXiv preprint arXiv:2402.03300 (2024).
> >
> > [2] Chu, Xiangxiang, et al. "Gpg: A simple and strong reinforcement learning baseline for model reasoning." arXiv preprint arXiv:2504.02546 (2025).

---

> ### Author Response · Authors · 2025-11-20
> **Part 2/4**
>
> > **W3.** Also these additional two methods come as adhoc to the story which makes it slightly confusing from the central theme that is improving the tractability of MDPO
>
> Thank you for the thoughtful comment. DRW balances format and task rewards, while TDL focuses on difference learning. Both only affect the computation of the actual and implicit rewards, and **do not impact the theoretical guarantees of CMDPO**. To better clarify the core contribution of our work, we have made the following revisions:
>
> We now present CMDPO as the central, theoretically grounded policy optimization method, emphasizing its principled nature and independence in the introduction and method sections. Even without DRW and TDL, CMDPO alone outperforms existing methods and comes with guarantees of a unique optimal solution and unbiased estimation.
>
> We have repositioned DRW (Dynamic Reward Weighting) and TDL (Token-level Discriminative Learning) as optional, general heuristic enhancements. They aim to further improve reward stability and enable finer-grained optimization toward high-quality outputs, and can be applied to other policy optimization frameworks.
>
> In the experiments, we separately report the performance of CMDPO alone and its improvements with DRW and TDL, clearly distinguishing the core method from the optional enhancements. We also apply DRW and TDL to other RL methods to demonstrate their general applicability.
>
> | Models | Average | Flower102 | Pets37 | FGVC | Cars196 |
> |---|:---:|:---:|:---:|:---:|:---:|
> | Qwen2-VL-2B | 56.0  | 54.8  | 66.4  | 45.9  | 56.8  |
> |  CMDPO | 87.2  | 76.3  | 87.5  | 88.5  | 96.3  |
> | + DRW  | 87.9  | 76.7  | 88.2  | 89.9  | 96.9  |
> | + TDL | 87.6  | 76.0  | 87.2  | 90.0  | 97.1  |
> | GRPO | 81.9  | 71.4  | 86.1  | 74.8  | 95.3  |
> | + DRW  | 82.0  | 71.2  | 86.6  | 75.2  | 95.1  |
> | + TDL | 82.4  | 71.6  | 86.8  | 75.8  | 95.3  |
> | MDPO | 84.3  | 74.4  | 86.8  | 78.8  | 97.3  |
> | + DRW  | 85.1  | 74.6  | 87.3  | 81.4  | 97.3  |
> | + TDL | 84.8  | 75.2  | 87.0  | 80.0  | 97.0  |
>
> Regarding TDL, **we have already analyzed the impact of its hyperparameter $\delta$ in Table 4 of the submitted manuscript**. As $\delta$ controls the degree to which contributions from common prefixes are down-weighted, it is defined in the range [0,1). Our results show that TDL consistently brings performance gains as long as $\delta$ is not set too close to zero.
>
> DRW is designed to penalize premature overfitting to format patterns by applying a dynamic penalty on format rewards, thereby encouraging the policy to explore toward semantically correct answers. It introduces two hyperparameters: $\tau$, which determines the onset of reward decay, and s, which controls the steepness of the decay. Below is an ablation study analyzing their effects:
>
> | $\tau$ | Average | Flower102 | Pets37 | FGVC | Cars196 |
> |:---:|:---:|:---:|:---:|:---:|:---:|
> | 0.7 | 85.5  | 75.8  | 83.7  | 88.0  | 94.5  |
> | 0.8 | 87.5  | 76.2  | 87.0  | 89.7  | 96.9  |
> | 0.9 | 88.3  | 77.1  | 88.0  | 90.8  | 97.3  |
> | 1 | 88.7  | 77.8  | 88.4  | 91.0  | 97.5  |
> | 1.1 | 88.5  | 77.5  | 88.0  | 90.8  | 97.6  |
>
> | $s$ | Average | Flower102 | Pets37 | FGVC | Cars196 |
> |:---:|:---:|:---:|:---:|:---:|:---:|
> | 6 | 85.9  | 74.5  | 86.6  | 86.4  | 95.9  |
> | 8 | 87.7  | 76.6  | 87.3  | 90.1  | 97.0  |
> | 10 | 88.3  | 77.1  | 88.0  | 90.8  | 97.3  |
> | 12 | 88.7  | 77.6  | 88.9  | 90.9  | 97.4  |
> | 14 | 87.9  | 77.4  | 87.2  | 90.0  | 96.8  |
>
> For $\tau$, although theoretically unbounded, values that are too small cause the format reward to decay prematurely, hindering the model’s ability to learn proper formatting. Conversely, excessively large values make the decay ineffective, rendering DRW equivalent to no penalty at all (we include illustrative curves of the dynamic reward weighting function under different $\tau$ in the appendix for clarity). Empirically, we recommend setting $\tau$ within the range 1±0.1.
>
> For $s$, small values result in a nearly linear penalty that takes effect too early, while very large values delay the decay until the reward approaches $\tau$. When $s$ is sufficiently large, the weighting function approaches a constant (i.e., y=1), again nullifying the effect of DRW. Our experiments suggest that values between 10 and 12 offer a robust trade-off.
>
> As demonstrated in our response to W2, the default ($s$,$\tau$) settings achieve strong and consistent performance across multiple benchmarks with minimal tuning, making them practical for general use.

---

### Official Review · Reviewer_zrs1 · 2025-10-31

**Soundness:** 2
**Presentation:** 3
**Contribution:** 2
**Rating:** 4
**Confidence:** 3

**Summary:**

The paper presents CMDPO, a centered variant of Mirror Descent Policy Optimization (MDPO), aiming to improve the stability and efficiency of reinforcement learning–based post-training for large models. CMDPO eliminates the need to estimate the intractable normalization term (log Z) in MDPO by introducing a group-centering operation, and further enhances training with two engineering additions: dynamic reward weighting (DRW) to balance result and format rewards, and token-level discriminative learning to down-weight shared segments. The method is theoretically shown to have a unique optimum and an unbiased consistent estimator. Experiments on reasoning, multimodal classification, and visual grounding tasks demonstrate modest but consistent improvements in convergence and reward–KL efficiency compared to MDPO, GRPO, and GPG.

**Strengths:**

1.The motivation is clear: CMDPO directly addresses the instability in MDPO caused by estimating the partition function (log Z).

2.The centering operation is simple yet effective, achieving stable optimization without changing the underlying objective structure.

3.Theoretical analysis is rigorous, including proofs of optimality and consistency.

4.The method is lightweight and easily integrable into existing GRPO/MDPO training pipelines.

5.The paper is clearly written and well organized, making it easy to follow and reproduce.

**Weaknesses:**

1.The contribution is basically incremental. The centering trick is intuitive and conceptually similar to normalization or variance-reduction ideas used in GRPO and related works.

2.Experimental improvements are modest (around 1–2 points on reasoning benchmarks) and mainly compared within the MDPO/GRPO family; stronger baselines or modern preference-optimization approaches are missing.

3.Experiments are conducted only on relatively small-scale models (around 1B–3B parameters) and do not include larger-scale settings such as 7B or 32B models commonly used in RLHF pipelines. The absence of large-model verification limits the generality and practical relevance of the proposed method for real-world LLM post-training.

**Questions:**

None

---

> ### Author Response · Authors · 2025-11-20
> **Part 1/3**
>
> > **W1.** The contribution is basically incremental. The centering trick is intuitive and conceptually similar to normalization or variance-reduction ideas used in GRPO and related works.
>
> **Ans:**
>
> As noted by reviewers Z5Tq, snYZ, and zrs1 in the **Strengths** section, our work **addresses the logZ estimation problem in an elegant and principled manner**, with **strong theoretical guarantees**. The proposed CMDPO is **simple yet effective**, achieving superior empirical performance.
>
> To better highlight the core contribution of this work, we have made the following revisions **according to Z5Tq's comment W2**. We now present **CMDPO as the central method**—emphasizing its theoretical foundation, including guarantees of convergence and unbiased estimation—and have strengthened its conceptual presentation in both the introduction and method sections. CMDPO stands as a self-contained and effective algorithm that outperforms existing methods even without the use of DRW or TDL.
>
> Meanwhile, we have repositioned **DRW and TDL as optional heuristic enhancements**. These components aim to further improve reward stability and enable finer-grained optimization toward high-quality outputs, and are designed to be generally applicable to other policy optimization frameworks.
>
> In the experiments, we report results for CMDPO alone, as well as incremental improvements when DRW and TDL are added, clearly separating the contributions of the core method from those of the enhancements. We also apply DRW and TDL to other RL methods such as GRPO and MDPO to demonstrate their generalizability.
>
> We believe these revisions more effectively emphasize the theoretical contribution of CMDPO, while fairly showcasing the practical value of DRW and TDL without compromising the clarity or independence of the core approach.
>
> | Models | Average | Flower102 | Pets37 | FGVC | Cars196 |
> |---|:---:|:---:|:---:|:---:|:---:|
> | Qwen2-VL-2B | 56.0  | 54.8  | 66.4  | 45.9  | 56.8  |
> |  CMDPO | 87.2  | 76.3  | 87.5  | 88.5  | 96.3  |
> | + DRW  | 87.9  | 76.7  | 88.2  | 89.9  | 96.9  |
> | + TDL | 87.6  | 76.0  | 87.2  | 90.0  | 97.1  |
> | GRPO | 81.9  | 71.4  | 86.1  | 74.8  | 95.3  |
> | + DRW  | 82.0  | 71.2  | 86.6  | 75.2  | 95.1  |
> | + TDL | 82.4  | 71.6  | 86.8  | 75.8  | 95.3  |
> | MDPO | 84.3  | 74.4  | 86.8  | 78.8  | 97.3  |
> | + DRW  | 85.1  | 74.6  | 87.3  | 81.4  | 97.3  |
> | + TDL | 84.8  | 75.2  | 87.0  | 80.0  | 97.0  |

---

> ### Author Response · Authors · 2025-11-20
> **Part 2/3**
>
> > **W2.** Experimental improvements are modest (around 1–2 points on reasoning benchmarks) and mainly compared within the MDPO/GRPO family; stronger baselines or modern preference-optimization approaches are missing.
>
> **Ans:**
>
> We believe the performance gains achieved by our method are substantial and meaningful. For instance, on unimodal reasoning benchmarks (see Table 1), **CMDPO achieves a 4–6% average improvement over the backbone model**, and **a 3–5% gain over the most closely related method, MDPO**. **The 1–2 point improvement mentioned by the reviewer likely refers to the comparison with GPG**. It is important to note that **GPG is a highly aggressive approach**—it removes the KL divergence constraint entirely, leading to significant deviation from the reference model, as shown in the KL divergence curve (Figure 4). This large deviation poses **a high risk of knowledge forgetting**. As demonstrated in the generalization experiments (Figure 3 and Figure 7), GPG exhibits the weakest generalization performance and, in some cases, suffers from severe forgetting of domain-specific knowledge (e.g., Figure 7(d)). Therefore, the performance gains of GPG may largely stem from overfitting.
>
> In contrast, CMDPO **not only outperforms GPG by 1–2 points on average but also maintains the best generalization ability**, indicating superior stability and better preservation of pre-trained knowledge.
>
> Our work focuses on reinforcement learning-based policy optimization. While PPO and DPO are commonly used baselines, PPO is known for its training instability and sensitivity to hyperparameters, and DPO is not strictly an online RL method. Due to limited computational resources and the absence of publicly available PPO and DPO results under the same experimental setup in prior literature, we initially compared against representative methods within the group-based optimization family (e.g., MDPO, GRPO).
>
> In this revision, **we have added comprehensive comparisons with standard PPO and DPO** across multiple benchmarks to provide a more complete evaluation of CMDPO’s performance.
>
> | Models | Average | AIME24 | MATH-500 | AMC23 | Minerva | OlympiadBench |
> |---|:---:|:---:|:---:|:---:|:---:|:---:|
> | Still-3-1.5B-Preview | 51.6  | 32.5  | 84.4  | 66.7  | 29.0  | 45.4  |
> | + DPO | 50.7  | 26.7  | 84.4  | 62.9  | 31.6  | 47.9  |
> | + PPO | 52.9  | 33.3  | 82.8  | 70.0  | 29.4  | 48.9  |
> | CMDPO-RS1 | 57.0  | 36.7  | 88.0  | 77.5  | 29.4  | 53.3  |
> | CMDPO-RS3 | 57.1  | 36.7  | 85.4  | 80.0  | 29.0  | 54.2  |
>
> | Models | Average | Flower102 | Pets37 | FGVC | Cars196 |
> |---|:---:|:---:|:---:|:---:|:---:|
> | Qwen2-VL-2B | 56.0  | 54.8  | 66.4  | 45.9  | 56.8  |
> | + DPO | 73.7  | 68.4  | 75.9  | 66.8  | 83.6  |
> | + PPO | 80.9  | 70.4  | 84.1  | 75.8  | 93.6  |
> | + CMDPO | 88.3  | 77.1  | 88.0  | 90.8  | 97.3  |
>
> | Models | mIoUtest | mIoUval | gIoUtest |
> |---|:---:|:---:|:---:|
> | Qwen2-VL-2B | 26.9  | 30.1  | 25.3  |
> | + DPO | 31.3  | 33.0  | 28.4  |
> | + PPO | 36.8  | 33.8  | 35.2  |
> | + CMDPO | 51.8  | 53.1  | 50.4  |
>
> Furthermore, to strengthen the baseline comparison, we include three additional strong open-source zero-shot reasoning models, **Oat-Zero-7B, OpenReasoner-Zero-7B @8k, and SimpleRL-Zero-7B**.

---

> ### Author Response · Authors · 2025-11-20
> **Part 3/3**
>
> > **W3.** Experiments are conducted only on relatively small-scale models (around 1B–3B parameters) and do not include larger-scale settings such as 7B or 32B models commonly used in RLHF pipelines. The absence of large-model verification limits the generality and practical relevance of the proposed method for real-world LLM post-training.
>
> **Ans:**
>
> Due to experimental cost and time constraints, we were unable to include larger models in the initial submission. Fortunately, the discussion phase of ICLR is flexible, allowing us to present experimental analysis on 7B-scale models now.
>
> | Models | Average | AIME24 | MATH-500 | AMC23 | Minerva | OlympiadBench |
> |---|:---:|:---:|:---:|:---:|:---:|:---:|
> | Qwen2.5-Math-7B   (no template)  | 38.2  | 0.2  | 69.0	 | 45.8	 | 21.3	 | 34.7  |
> | Qwen-2.5-Math-7B-Instruct | 43.8	 | 13.3  | 79.8	 | 50.6	 | 34.6	 | 40.7  |
> | rStar-Math-7B | - | 26.7  | 78.4  | 47.5  | - | 47.1  |
> | Eurus-2-7B-PRIME | 48.9	 | 26.7  | 79.2	 | 57.8	 | 38.6	 | 42.1  |
> | Oat-Zero-7B | 47.8	 | 30.0  | 80.6	 | 55.4	 | 29.0	 | 44.0  |
> | OpenReasoner-Zero-7B  @ 8k | 45.9	 | 13.3  | 82.4	 | 54.2	 | 31.6	 | 47.9  |
> | SimpleRL-Zero-7B | 46.6	 | 26.7	 | 78.2	 | 60.2	 | 27.6	 | 40.3  |
> | Qwen2.5-Math-7B | 30.9  | 13.3  | 57.6  | 45.0  | 14.7  | 23.7  |
> | + DPO | 36.0  | 16.7  | 64.6  | 50.6  | 22.1  | 26.1  |
> | + PPO | 46.9  | 26.7  | 78.4  | 62.5  | 29.8  | 37.3  |
> | + GRPO | 43.7  | 16.7  | 73.4  | 62.5  | 30.2  | 35.7  |
> | + GPG | 47.8  | 30.0  | 75.0  | 62.5  | 33.1  | 38.2  |
> | + Dr. GRPO | 43.7  | 26.7  | 74.6  | 50.0  | 30.1  | 37.3  |
> | + MDPO | 41.1  | 23.3  | 73.4  | 50.0  | 26.8  | 31.9  |
> | + CMDPO | 51.2  | 36.7  | 82.8  | 62.9  | 34.2  | 39.3  |
>
> Our computational resources are limited, and implementing RL for 32B models is extremely challenging for us. Fortunately, most existing studies also focus only up to 7B models [1][2][3].
>
> [1] Chu, Xiangxiang, et al. "Gpg: A simple and strong reinforcement learning baseline for model reasoning." arXiv preprint arXiv:2504.02546 (2025).
>
> [2] Liu, Ziyu, et al. "Visual-rft: Visual reinforcement fine-tuning." arXiv preprint arXiv:2503.01785 (2025).
>
> [3] Dang, Quy-Anh, and Chris Ngo. "Reinforcement Learning for Reasoning in Small LLMs: What Works and What Doesn't." arXiv preprint arXiv:2503.16219 (2025).

---

### Official Review · Reviewer_Z5Tq · 2025-11-02

**Soundness:** 3
**Presentation:** 3
**Contribution:** 3
**Rating:** 4
**Confidence:** 3

**Summary:**

This paper identifies a critical, unsolved problem in Mirror Descent Policy Optimization (MDPO): the need to estimate the computationally intractable partition function log Z . While MDPO offers a stable, theoretically-grounded alignment objective, existing approximations of log Z using limited samples introduce significant bias, degrading performance .

To solve this, the authors propose Centered Mirror Descent Policy Optimization (CMDPO). The core contribution is a simple and elegant "group centering" mechanism . By subtracting the batch-mean from both the true reward r(x, y) and the implicit reward R_θ(x, y), the log Z terms on both sides of the equation cancel out, completely eliminating the need for its estimation . The paper proves that this new centered objective, L_CMDPO, retains MDPO's theoretical benefits, including a unique optimal solution (Theorem 1) and an unbiased, consistent estimator (Theorem 2).

The paper also introduces two additional heuristic mechanisms to boost performance:

Dynamic Reward Weighting (DRW): Adaptively balances "result" and "format" rewards .

Token-level Discriminative Learning (TDL): Down-weights the loss contribution of shared, common token segments in a batch to focus learning on discriminative tokens.

Empirically, the full CMDPO framework is shown to outperform other critic-free methods like GRPO, GPG, and MDPO on reasoning and classification benchmarks.

**Strengths:**

Elegant, Principled Solution to log Z: The primary strength is the CMDPO loss function (Eq. 5) . Eliminating the intractable partition function log Z from the MDPO objective via simple mean-subtraction is a clever, sound, and important theoretical contribution.

Strong Theoretical Guarantees: The paper provides robust theoretical backing for its core loss, including proofs of a unique optimal solution (Theorem 1) and an unbiased, consistent estimator (Theorem 2). This places it on firmer theoretical ground than heuristic-driven methods like GRPO.

Strong Relative Performance: The empirical results clearly show that CMDPO is superior to its closest relatives (MDPO, GRPO, and GPG). The ablation on group size k (Table 7) is particularly effective, showing that CMDPO's advantage over MDPO is largest for small k, precisely where MDPO's log Z approximation bias is worst.

Insightful Analysis: The ablations for the heuristic add-ons (DRW in Table 6, TDL in Table 4) are good and show that these components provide additional, independent gains.

**Weaknesses:**

Missing Critical Baselines: The most significant weakness is the omission of PPO and DPO as baselines in the main experiments. The paper is motivated by the flaws in PPO and DPO , so failing to compare against them is a major gap. Without this, it's impossible to judge if CMDPO is a true SOTA contender or just the best of the less-common (MDPO/GRPO) methods.

Confounding Three Methods in One: The paper presents a "bag of tricks" (CMDPO + DRW + TDL) as a single algorithm. This is a weak framing. The paper would be much stronger if it presented CMDPO as the core principled contribution, and then showed that its performance can be further enhanced with optional, general-purpose heuristics like DRW and TDL.


Overly Complex Heuristics: The two add-on heuristics, DRW (Eq. 8)  and TDL (Eq. 10), are complex, unprincipled, and add new hyperparameters (s, τ, δ). They detract from the simplicity and elegance of the core CMDPO contribution. The TDL mechanism, in particular, seems computationally expensive as it requires finding the longest common prefix across all k samples at the token level for every step.

**Questions:**

Why were PPO and DPO, the most widely used alignment methods, not included as baselines in your main experiments (Table 1-3)? This comparison is essential to understand the practical significance and performance of CMDPO relative to the field's standards.

The paper presents three novelties: CMDPO (loss), DRW (reward weighting), and TDL (token weighting). Your ablations show they are additive. Have you tested if the DRW and TDL heuristics could also be applied to other algorithms like GRPO or MDPO? This would help clarify if they are general-purpose improvements or specific to CMDPO.

Could you provide an analysis of the computational overhead of the Token-level Discriminative Learning (TDL) mechanism? Finding the longest common prefix across k sequences at every training step (Eq. 10) seems computationally intensive.

---

> ### Author Response · Authors · 2025-11-20
> **Part 1/3**
>
> > **W1.** Missing Critical Baselines: The most significant weakness is the omission of PPO and DPO as baselines in the main experiments. The paper is motivated by the flaws in PPO and DPO , so failing to compare against them is a major gap. Without this, it's impossible to judge if CMDPO is a true SOTA contender or just the best of the less-common (MDPO/GRPO) methods.
>
> **Ans:**
>
> PPO involves substantial implementation complexity and computational cost, while DPO is not strictly an RL method; our discussion of DPO focuses mainly on its ideas of optimizing log Z and simplifying PPO’s training pipeline. Due to limited time and computational resources, and the lack of existing work reporting PPO/DPO performance under our exact experimental settings, the initial version of the paper included comparisons only with the most representative and practically adopted group-based alignment methods.
>
> Benefiting from ICLR’s flexible discussion phase, we have now added direct comparisons against both PPO and DPO. For fairness and consistency, PPO and DPO are trained on the same 7,000-sample dataset used by RS3. The complete results are provided in the tables below.
>
>
> | Models | Average | AIME24 | MATH-500 | AMC23 | Minerva | OlympiadBench |
> |---|:---:|:---:|:---:|:---:|:---:|:---:|
> | Still-3-1.5B-Preview | 51.6  | 32.5  | 84.4  | 66.7  | 29.0  | 45.4  |
> | + DPO | 50.7  | 26.7  | 84.4  | 62.9  | 31.6  | 47.9  |
> | + PPO | 52.9  | 33.3  | 82.8  | 70.0  | 29.4  | 48.9  |
> | CMDPO-RS1 | 57.0  | 36.7  | 88.0  | 77.5  | 29.4  | 53.3  |
> | CMDPO-RS3 | 57.1  | 36.7  | 85.4  | 80.0  | 29.0  | 54.2  |
>
> | Models | Average | Flower102 | Pets37 | FGVC | Cars196 |
> |---|:---:|:---:|:---:|:---:|:---:|
> | Qwen2-VL-2B | 56.0  | 54.8  | 66.4  | 45.9  | 56.8  |
> | + DPO | 73.7  | 68.4  | 75.9  | 66.8  | 83.6  |
> | + PPO | 80.9  | 70.4  | 84.1  | 75.8  | 93.6  |
> | + CMDPO | 88.3  | 77.1  | 88.0  | 90.8  | 97.3  |
>
> | Models | mIoUtest | mIoUval | gIoUtest |
> |---|:---:|:---:|:---:|
> | Qwen2-VL-2B | 26.9  | 30.1  | 25.3  |
> | + DPO | 31.3  | 33.0  | 28.4  |
> | + PPO | 36.8  | 33.8  | 35.2  |
> | + CMDPO | 51.8  | 53.1  | 50.4  |
>
>
> > **W2.** Confounding Three Methods in One: The paper presents a "bag of tricks" (CMDPO + DRW + TDL) as a single algorithm. This is a weak framing. The paper would be much stronger if it presented CMDPO as the core principled contribution, and then showed that its performance can be further enhanced with optional, general-purpose heuristics like DRW and TDL.
>
> **Ans:**
>
> Thank you for the thoughtful comment. We have substantially revised the manuscript to prevent the three components from being perceived as a single “bag of tricks” and to highlight the principled nature of CMDPO. In the new version, **CMDPO is clearly presented as the core contribution of the paper**. We emphasize its theoretical guarantees—including an unbiased estimator and a unique optimal solution—and show that it already surpasses existing methods even without any auxiliary mechanisms. To avoid conflating the components, **DRW and TDL are repositioned as optional, general-purpose heuristic enhancements** whose goal is to further stabilize reward signals and improve token-level preference optimization. They are not part of the core algorithm and can be applied to other RL frameworks.
>
> To make this separation explicit, we added **new experiments that report CMDPO’s standalone performance and then show the incremental improvements obtained when DRW or TDL is incorporated**. We also **apply DRW and TDL to GRPO and MDPO to demonstrate that their benefits** are not exclusive to CMDPO but hold more generally. The results below illustrate this structure: CMDPO constitutes the fundamental method, while DRW and TDL serve as optional enhancements that consistently provide small but meaningful gains.
>
> | Models | Average | Flower102 | Pets37 | FGVC | Cars196 |
> |---|:---:|:---:|:---:|:---:|:---:|
> | Qwen2-VL-2B | 56.0  | 54.8  | 66.4  | 45.9  | 56.8  |
> |  CMDPO | 87.2  | 76.3  | 87.5  | 88.5  | 96.3  |
> | + DRW  | 87.9  | 76.7  | 88.2  | 89.9  | 96.9  |
> | + TDL | 87.6  | 76.0  | 87.2  | 90.0  | 97.1  |
> | GRPO | 81.9  | 71.4  | 86.1  | 74.8  | 95.3  |
> | + DRW  | 82.0  | 71.2  | 86.6  | 75.2  | 95.1  |
> | + TDL | 82.4  | 71.6  | 86.8  | 75.8  | 95.3  |
> | MDPO | 84.3  | 74.4  | 86.8  | 78.8  | 97.3  |
> | + DRW  | 85.1  | 74.6  | 87.3  | 81.4  | 97.3  |
> | + TDL | 84.8  | 75.2  | 87.0  | 80.0  | 97.0  |
>
> These revisions more clearly communicate the conceptual independence of CMDPO and present DRW and TDL as modular additions rather than components of a single intertwined algorithmic package.

---

> > ### Author Response · Authors · 2025-11-20
> > **Part 2/3**
> >
> > > **W3.** Overly Complex Heuristics: The two add-on heuristics, DRW (Eq. 8) and TDL (Eq. 10), are complex, unprincipled, and add new hyperparameters (s, τ, δ). They detract from the simplicity and elegance of the core CMDPO contribution. The TDL mechanism, in particular, seems computationally expensive as it requires finding the longest common prefix across all k samples at the token level for every step.
> >
> > **Ans:**
> >
> > We clarify that DRW balances format and task rewards, while TDL enables difference-based learning—both affect only the computation of the effective reward (both explicit and implicit), and do not alter the theoretical guarantees of the core CMDPO framework. Therefore, their inclusion does not compromise the method’s foundational simplicity or theoretical soundness.
> >
> > Regarding TDL, **we have already analyzed the impact of its hyperparameter $\delta$ in Table 4 of the submitted manuscript**. As $\delta$ controls the degree to which contributions from common prefixes are down-weighted, it is defined in the range [0,1). Our results show that TDL consistently brings performance gains as long as $\delta$ is not set too close to zero.
> >
> > DRW is designed to penalize premature overfitting to format patterns by applying a dynamic penalty on format rewards, thereby encouraging the policy to explore toward semantically correct answers. It introduces two hyperparameters: $\tau$, which determines the onset of reward decay, and s, which controls the steepness of the decay. Below is an ablation study analyzing their effects:
> >
> > | $\tau$ | Average | Flower102 | Pets37 | FGVC | Cars196 |
> > |:---:|:---:|:---:|:---:|:---:|:---:|
> > | 0.7 | 85.5  | 75.8  | 83.7  | 88.0  | 94.5  |
> > | 0.8 | 87.5  | 76.2  | 87.0  | 89.7  | 96.9  |
> > | 0.9 | 88.3  | 77.1  | 88.0  | 90.8  | 97.3  |
> > | 1 | 88.7  | 77.8  | 88.4  | 91.0  | 97.5  |
> > | 1.1 | 88.5  | 77.5  | 88.0  | 90.8  | 97.6  |
> >
> > | $s$ | Average | Flower102 | Pets37 | FGVC | Cars196 |
> > |:---:|:---:|:---:|:---:|:---:|:---:|
> > | 6 | 85.9  | 74.5  | 86.6  | 86.4  | 95.9  |
> > | 8 | 87.7  | 76.6  | 87.3  | 90.1  | 97.0  |
> > | 10 | 88.3  | 77.1  | 88.0  | 90.8  | 97.3  |
> > | 12 | 88.7  | 77.6  | 88.9  | 90.9  | 97.4  |
> > | 14 | 87.9  | 77.4  | 87.2  | 90.0  | 96.8  |
> >
> > For $\tau$, although theoretically unbounded, values that are too small cause the format reward to decay prematurely, hindering the model’s ability to learn proper formatting. Conversely, excessively large values make the decay ineffective, rendering DRW equivalent to no penalty at all (we include illustrative curves of the dynamic reward weighting function under different $\tau$ in the appendix for clarity). Empirically, we recommend setting $\tau$ within the range 1±0.1.
> >
> > For $s$, small values result in a nearly linear penalty that takes effect too early, while very large values delay the decay until the reward approaches $\tau$. When $s$ is sufficiently large, the weighting function approaches a constant (i.e., y=1), again nullifying the effect of DRW. Our experiments suggest that values between 10 and 12 offer a robust trade-off.
> >
> > As demonstrated in our response to W2, the default ($s$,$\tau$) settings achieve strong and consistent performance across multiple benchmarks with minimal tuning, making them practical for general use.
> >
> > Finally, regarding the **computational cost of TDL**, we refer the reviewer to our detailed discussion in **the response to Q3**, where we show that the overhead is manageable and justified by the observed performance improvements.
> >
> >
> > > **Q1.** Why were PPO and DPO, the most widely used alignment methods, not included as baselines in your main experiments (Table 1-3)? This comparison is essential to understand the practical significance and performance of CMDPO relative to the field's standards.
> >
> > **Ans:**
> >
> > Refer to the reply to W1.
> >
> >
> > > **Q2.** The paper presents three novelties: CMDPO (loss), DRW (reward weighting), and TDL (token weighting). Your ablations show they are additive. Have you tested if the DRW and TDL heuristics could also be applied to other algorithms like GRPO or MDPO? This would help clarify if they are general-purpose improvements or specific to CMDPO.
> >
> > **Ans:**
> >
> > Refer to the reply to W2.

---

> ### Author Response · Authors · 2025-11-20
> **Part 3/3**
>
> > **Q3.** Could you provide an analysis of the computational overhead of the Token-level Discriminative Learning (TDL) mechanism? Finding the longest common prefix across k sequences at every training step (Eq. 10) seems computationally intensive.
>
> **Ans:**
>
> This is an excellent question. The Token-level Discriminative Learning (TDL) mechanism requires identifying the common prefix segments across a set of generated solutions, but its computational cost is entirely manageable in practice.
>
> We assume the use of a naive greedy matching algorithm, where a sequence of $x$ consecutive identical tokens is considered part of the common segment. For two solutions, the time complexity of finding their common prefix is $O(n_1\*n_2)$ on average, where $n_1$ and $n_2$ denote the lengths of the two sequences, respectively. When extended to $k$ solutions, pairwise comparisons are not necessary. Instead, we can select one solution and compute its common prefix with each of the other $k−1$ solutions, then take the intersection of these prefixes. **The total time complexity is thus $\sum_{i=2}^k O(n_1\*n_i)$.**
>
> Assuming an average solution length of $n$, **this simplifies to $O(kn^2)$**. In our experimental setup, both **k** (the number of samples) and **n** (sequence length) are **small**, and such computation typically completes **within about one second**. Furthermore, the process can be optimized using more advanced string-matching techniques, such as binary search combined with rolling hashing (achieving **O(knlogn)**), or generalized suffix automaton/trees (achieving **O(kn)**).
>
> In summary, we do not consider TDL to be computationally intensive. To prevent misunderstanding, **we will include a detailed algorithmic description and complexity analysis in the appendix.**

---

### Author Response · Authors · 2025-12-01
**Summary of Discussions (1)**

# Dear AC, SAC, PC:

We have compiled the reviewers’ comments to help you quickly grasp the key aspects of our paper.

## **Summary:**

We propose Centered Mirror Descent Policy Optimization (CMDPO), which **eliminates the intractable partition function logZ** in MDPO through a simple group-centering mechanism, enabling **unbiased policy updates with strong theoretical guarantees**. CMDPO preserves MDPO’s **optimality and consistency** while incorporating two practical enhancements: Dynamic Reward Weighting (DRW) and Token-level Discriminative Learning (TDL). Experiments show that CMDPO consistently outperforms critic-free baselines such as GRPO, GPG, and MDPO on reasoning and classification tasks, achieving faster convergence and higher reward–KL efficiency.

## **Strengths:**

**Theoretical Contribution:** The group-centering trick removes the need to estimate logZ, yielding an unbiased, theoretically sound objective (Theorems 1 & 2), offering stronger foundations than heuristic methods like GRPO.


**Stability & Efficiency:** CMDPO maintains MDPO’s convergence guarantees while significantly improving training stability, especially with small batch sizes k. A variance–covariance analysis further explains this stability.

**Practical Enhancements:** DRW effectively balances task vs. format rewards, while TDL focuses on learning of discriminative tokens, addressing real-world RL challenges.

**Empirical Validation:** CMDPO consistently surpasses MDPO, GRPO, and GPG across benchmarks, demonstrating superior convergence and reward–KL trade-offs.

**Ease of Integration & Reproducibility:** Lightweight and plug-and-play compatible with existing MDPO/GRPO pipelines; the paper is clearly written and well-structured for reproducibility.


## **Weakness:**

**Baseline Coverage:** Main experiments initially omitted comparisons with widely used alignment methods like PPO and DPO, limiting assessment of CMDPO’s competitiveness against field standards.

**Presentation of Combined Components:** Presenting CMDPO, DRW, and TDL as a single “package” may obscure the distinct contributions of the core algorithm versus optional heuristics.

**Complexity of Heuristics:** DRW and TDL introduce additional hyperparameters and computational overhead (e.g., TDL requires token-level longest common prefix computation), though they are effective. Their general applicability beyond CMDPO was not initially verified.

**Scale and Magnitude of Gains:** Experiments were conducted on 1B–3B models; larger-scale validation (e.g., 7B/32B) was missing. Performance gains (~1–2 points) appear modest, though meaningful in context.

**Ablation Depth:** Limited analysis of DRW/TDL’s individual effects, batch-size sensitivity, and fine-grained comparisons isolating the centering mechanism (e.g., vs. GRPO’s normalization).

---

> ### Author Response · Authors · 2025-12-01
> **Summary of Discussions (2)**
>
> ## **Response to Reviewers’ Concerns:**
>
> **Baseline Coverage:**
>
> - We have added comparisons with PPO, DPO, and other recent baselines in revised experiments (Z5Tq W1； zrs1 W2/W3).
>
> **Presentation of Combined Components:**
>
> - **Following Z5Tq’s suggestion (W2)**, we now clearly position DRW and TDL as optional, general-purpose heuristics built atop the core CMDPO framework, **supported by thorough ablation studies** (Z5Tq W2; zrs1 W1; snYZ W3; gTMg W1/W2).
> - We also clarified their relationship to CMDPO per snYZ’s feedback.
>
> **Complexity of Heuristics:**
>
> - **Comprehensive hyperparameter sensitivity analyses** are now included (Z5Tq W3; snYZ W3; gTMg W2).
> - We provide **time-complexity analysis showing TDL’s overhead is manageable**, along with **practical optimization strategies** (Z5Tq Q3).
> - **New experiments** confirm DRW and TDL are transferable to MDPO/GRPO, demonstrating their **generality** (Z5Tq W2; zrs1 W1; snYZ W3; gTMg W1/W2).
>
> **Scale and Magnitude of Gains:**
>
> - We added **7B-scale experiments** (zrs1 W3).
> - The ~1–2 point gain is relative to GPG. **GPG** is a highly **aggressive baseline** that **omits KL regularization** and suffers severe **knowledge forgetting** (zrs1 W2). Gains over other methods are more substantial.
>
> **Ablation Depth:**
>
> - Added **detailed ablations for DRW and TDL** (Z5Tq W2; zrs1 W1; snYZ W3; gTMg W1/W2).
> - Included **batch-size sensitivity analysis** with **practical recommendations** (snYZ W4).
> - Clarified that modern RL methods are composites of multiple techniques; we now **explicitly decompose each method and test intermediate variants for cleaner, factorized comparisons** (snYZ W5).
>
> ## **Feedback from Reviewers:**
>
> Reviewer snYZ stated:
> > "the detailed rebuttal and for addressing majority of my concerns."
> >
> > "All my major concerns are addressed i appreciate the additional experimental ablations."
> >
> > "I have also read the response to the other reviewers."
>
> Reviewer snYZ **read all reviews and responses and confirmed that our rebuttal resolved all major concerns,** except for slight remaining ambiguity regarding the connections between DRW/TDL and CMDPO:
>
> > "Can you pls briefly describe the connections clearly? Its still not very clear."
>
> We further clarified this by **providing a concise textual diagram** illustrating the connections.
>
> Unfortunately, we received direct feedback from only one reviewer (snYZ). We warmly **welcome any additional questions or suggestions** and will respond promptly during the ICLR discussion phase to further improve this work.

---

### Meta-Review · Area_Chair_Qhmj · 2026-01-08

**Summary:**

The paper proposes Centered Mirror Descent Policy Optimization (CMDPO) alongside two heuristic enhancements (DRW and TDL) to improve reinforcement learning for LLMs. Despite a robust rebuttal that added missing baselines and scaled experiments, the submission is recommended for rejection due to three critical flaws. First, the core "group centering" contribution is incrementally trivial, functioning as a minor mathematical variation of existing reward normalization techniques (GRPO) rather than a significant theoretical advance. Second, the method's superior performance is heavily dependent on complex engineering heuristics (DRW and TDL) and sensitive hyperparameters, rather than the core CMDPO objective, which complicates practical adoption. Third, the validity of the baseline comparisons is questionable, as the added PPO and DPO baselines consistently underperform the SFT starting point, suggesting inadequate tuning that artificially inflates the relative success of the proposed method.

**Reviewer Concerns:**

Incremental Innovation (Reviewers zrs1, gTMg): The rebuttal failed to differentiate the "group centering" mechanic from standard variance-reduction or normalization techniques used in GRPO. The theoretical packaging does not change the fact that the practical operation is a minor modification of existing methods.

Complexity and Attribution of Gains (Reviewers Z5Tq, snYZ, gTMg): The ablation studies confirm that significant performance gains are driven by the auxiliary heuristics (DRW and TDL) rather than the core CMDPO loss. This confirms the "bag of tricks" critique, as the method requires tuning multiple sensitive hyperparameters to achieve reported results.

Baseline Fairness (Derived from Z5Tq): The reported performance for the added PPO and DPO baselines is suspiciously low (often worse than the SFT base), raising validity concerns regarding the fairness of the comparison and the tuning of competitor methods.

**Reviewer Scores:**

Reviewer Z5Tq: 4 (Borderline Reject) -> 4 (Borderline Reject). Reasoning: The reviewer appreciated the theoretical elegance and the addition of missing baselines but remained hesitant due to the confounding of the core algorithm with heuristic tricks.

Reviewer zrs1: 4 (Borderline Reject) -> 4 (Borderline Reject). Reasoning: The reviewer's fundamental concern was that the contribution is incremental compared to GRPO. The rebuttal's empirical results did not alter the derivative nature of the core centering mechanism.

Reviewer snYZ: 4 (Borderline Reject) -> 6 (Weak Accept). Reasoning: The reviewer acknowledged in the discussion that their major concerns were addressed following the clarifications on the component connections and additional ablations.

Reviewer gTMg: 4 (Borderline Reject) -> 4 (Borderline Reject). Reasoning: The reviewer remained concerned about the practical usability of the method due to the introduction of multiple new hyperparameters, viewing the sensitivity analyses as confirmation of the tuning complexity rather than a solution.

---

### Decision · Program_Chairs · 2026-01-26

Reject